# Early warning and proactive control strategies for power blackouts caused by gas network malfunctions

Fengshuo Yu [1], Qinglai Guo [1], Jianzhong Wu [2], Zheng Qiao[1] & Hongbin Sun [1,3] ✉

There is growing consensus that gas-fired generators will play a crucial role during the transition to net-zero energy systems, both as an alternative to coal-fired generators and as a flexibility service provider for power systems. However, malfunctions of gas networks have caused several large-scale power blackouts. The transition from coal and oil to gas fuels significantly increases the interdependence between gas networks and electric power systems, raising the risks of more frequent and widespread power blackouts due to the malfunction of gas networks. In a coupled gas–electricity system, the identification and transmission of gas network malfunction information, followed by the redispatch of electric power generation, occur notably faster than the propagation and escalation of the malfunction itself, e.g., significantly diminished pressure. On this basis, we propose a gas-electric early warning system that can reduce the negative impacts of gas network malfunctions on the power system. A proactive control strategy of the power system is also formulated based on the early warning indicators. The effectiveness of this method is demonstrated via case studies of a real coupled gas–electricity system in China.

For electric power systems, it has become a common trend to gradually eliminate their dependence on fossil fuels and turn to low-carbon energy sources, in particular renewables such as wind and photovoltaics[1,2]. However, limited by the physical characteristic of power transmission and the intermittency of renewable generation, the development of conventional synchronous generators with sufficient capacity are still required that can be freely adjusted to maintain the balance between generation and the load of power systems[3,4]. Among all types of conventional generators, gas turbines have attracted public attention due to their superior performance and lower prices[5]. In contrast to coal- and oil-fired generators, gas turbines emit lower levels of carbon and pollutants and exhibit higher ramp rates which facilitates the integration of variable renewable resources[6–8]. Many countries have built or are building large-scale gas networks and gas-fired power generators to reduce their carbon emission intensity[9], as shown in Fig. 1.

Moving towards carbon neutrality, although the use of natural gas as a fossil fuel will eventually be reduced, the gas infrastructure will be repurposed as an important energy carrier of alternative fuels (e.g., green hydrogen) with storage and transmission functionality[10,11]. Although the share of hydrogen production from the electrolysis of water is not as high as that from fossil fuels at present, the former is shown to be the best way to obtain large amounts of sustainable hydrogen with neither pollutants nor carbon emission[12,13]. Moreover, transporting these alternative fuels through established gas networks has proved to be a more economical way than trucking or constructing new pipelines[14]. In early 2021, Shell, Mitsubishi Heavy Industries, Vattenwall and Wärme Hamburge signed a letter of intent to develop renewable hydrogen production in Hamburg, where existing gas networks would be expanded to accommodate the transport of hydrogen[15]. In the same year, an initiative called HyBlend was launched

[1]Department of Electrical Engineering, Tsinghua University, Beijing, China. [2]School of Engineering, Cardiff University, Cardiff, UK. [3]College of Electrical and Power Engineering, Taiyuan University of Technology, Taiyuan, China. ✉e-mail: shb@tsinghua.edu.cn

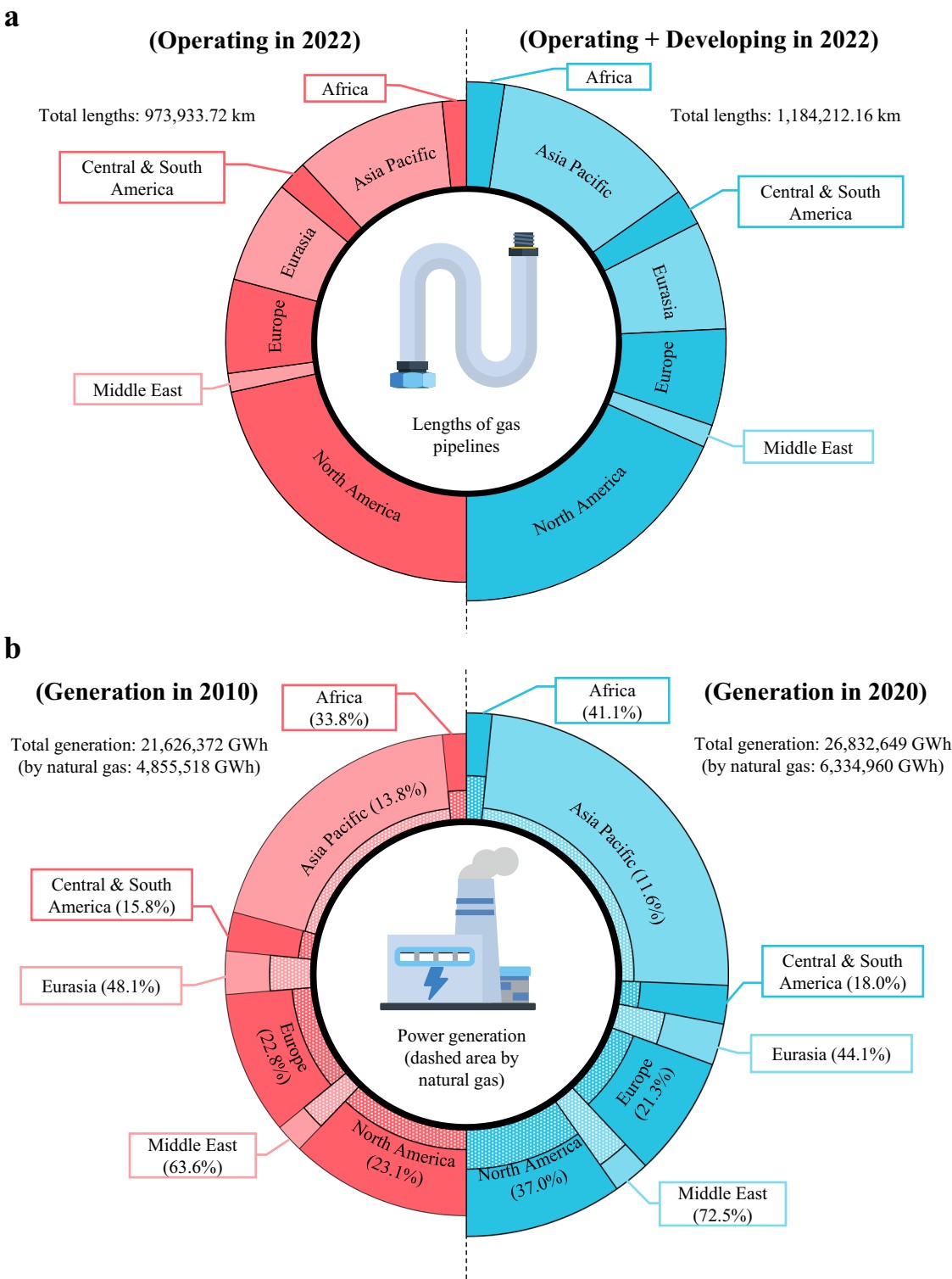

**Fig. 1 | Development of natural gas utilities worldwide. a** Kilometres of gas pipelines worldwide at the end of 2022. The size of each sector on the left indicates the operating length of gas pipelines in a region, while that on the right denotes the length of gas pipelines in development, reflecting the growth momentum of gas infrastructure. **b** Global power generation and the proportion of natural gas in 2010 and 2020. The size of each sector indicates the annual power generation in a region, while the dashed area denotes the proportion generated by natural gas. Data are retrieved from the literature[46,47]. Source data are provided as a Source Data file.

by the Office of Energy Efficiency and Renewable Energy (EERE) of the US, including R&D conducted by four national laboratories[16]. In July 2023, UK Research and Innovation (UKRI) declared a funding allocation of £95.3 million to push forward 10 large-scale demonstration projects, aiming to investigate the feasibility of blending green hydrogen into existing gas networks[17]. In this way, when the generation

capacity of renewables surpasses the load demand, the surplus power can be converted into green hydrogen and injected into the gas system; when the renewable generation falls short of the load demand, the power deficit can be supplemented through gas turbines.

However, compared to carbon emission reduction, the security of energy systems has not attracted enough attention from the public.

**Table 1 | Comparison of passive control and proactive control by the EPCC**

| Attribute | Passive control | Proactive control |
|---|---|---|
| Information used | Electric power system | Gas system and electric power system |
| When to act | When cascading failure propagates to the power system (slow response) | When a major failure occurs in the gas system (fast response) |
| Control effect | Extreme control measures, such as load shedding to offset the power deficit | Moderate control strategies of generator regulation to mitigate the power deficit |
| Impact on livelihood | Potentially very high | Avoided or significantly reduced |

With the expansion of gas networks, gas system malfunctions are now more likely to evolve into electric power outages. In February 2021, large-scale blackouts occurred in the south-central United States due to extremely cold weather[18]. Between February 8th and 20th, a gas generating capacity of 106,585 MW encountered unplanned outages or derates, primarily stemming from issues in the gas system[19]. Despite there were advance notifications to impacted gas turbines, the power system had to resort to load shedding in response to the crisis. In fact, similar incidents took place in the United States in 1983, 1989, 2003, 2004, 2006, 2008, 2009, 2010, and 2011 due to adverse weather conditions[20]. Additional factors, such as pipeline leakages and illegal operations may also lead to gas-electric cascading failures, as exemplified in California (2015)[21] and Taiwan (2017)[22]. In short, power outages caused by gas network malfunctions are presently widespread and are anticipated to pose an increasing risk in the near future. Firstly, the transition from coal and oil to natural gas increases the dependence of electric power systems on gas networks, and even the latter is gradually replaced by decarbonized alternative fuels, gas networks persist. Secondly, an increasing number of studies indicate that extreme weather events have become more frequent than before[23,24], which may not only directly do harm to power systems but also threaten them by affecting the transportation of gas fuel[25]. Thirdly, global geopolitical tension amplifies the likelihood of human factors impacting the security of gas supplies, such as the Nord Stream gas pipeline explosion[26].

So far, some research institutions, such as LANL, NREL, and EMPA, have put forward theories related to energy security or resilience from different angles to assess or enhance the capacity of energy systems to sustain normal energy supply amidst disturbances. Grouped by disturbance sources, these theories can be classified as dealing with internal disruptions (e.g., gas leakage[27]) or external disruptions (e.g., extreme weather events[28,29], cyber attacks[30]). Organized chronologically, they focus on the ability of energy systems to prepare for, absorb, adapt to, and recover from disruptive events before and after the disturbance[31,32]. In addition to traditional research on single energy resources, the exploration on the comprehensive resilience of integrated energy systems has also made breakthroughs (e.g., hurricane analysis[33], disaster avoidance[34]). However, most of the studies play a limited role in the prevention of gas-electric cascading failures for the lack of real-time coordination between the two entities. When the gas network malfunction occurs, there is a temporal misalignment of the failure concerning the two energy systems: the failure has already transpired in the gas system, while the power system is still on the verge of the failure. Current theories about energy resilience are less prone to utilize this propagation delay in the medium, which has actually been demonstrated by the earthquake early warning system in the area of geology and disaster response[35,36].

During a gas-electric cascading failure, the gas stored in pipelines (line pack) becomes a buffer for pressure variations, which typically makes the time required for the propagation and escalation of the malfunction much longer than the time needed for the electric power control centre (EPCC) to acquire failure information from the gas system. Meanwhile, the power system can swiftly adapt its operational state, offering the potential to proactively mitigate the gas-electric cascading failure. These facts inspired us to develop the *gas-electric early warning system* described in this paper. Compared to other security control methods, the gas-electric early warning system makes critical use of the delay between the gas network malfunction and its impact on gas turbines. Once the early warning system is introduced, the EPCC can employ global *proactive control* before the failure reaches the power system, rather than *passive control* after the electric failure occurs, as indicated in Table 1. In the former scenario, the impacts of gas network malfunctions on the electric power system will be significantly reduced relative to the scenario without the early warning system.

## Results
### Early warning of the gas system
Early warning is defined as the perception of an impending danger. A practical early warning system encompasses modules of sensing, analysis, decision-making, etc., intending to provide the management entity with more response time to minimize the losses[37]. For example, in the area of disaster response, predicting the precise timing of a tsunami is challenging, but there exists an obvious correlation between a coastal earthquake and a tsunami[36]. This correlation involves a time interval, which exceeds the reaction time required for the public to prepare for the tsunami, making the tsunami early warning system practicable.

In coupled gas-electricity systems, there is such a correlation between certain major failures of the gas network and cascading failures of the electric power system. When the information is limited to the power system, it is hard for the EPCC to predict unplanned outages of gas-fired generators. However, when the object of concern expands to the coupled gas-electricity system, there is a clear sequential correlation between gas supply interruptions and their impact on gas turbines due to insufficient inlet pressure. For gas-fired generators, when the inlet pressure drops to a certain extent, their maximum power generation is greatly reduced for safety purposes; as the inlet pressure drops further, they must be shut down to prevent equipment anomalies. In fact, pressure is one of the most crucial state variables for the normal operation of gas facilities, including valves, compressors, and gas loads. Its deviation commonly serves as a reason for cascading failures within the gas system, just as deviations of voltage lead to cascading failures within the power system. Nevertheless, a comparison between the pipeline equations of gas systems and the telegraph equations of power systems reveals a notable distinction. Gas systems demonstrate a stronger capacity to maintain constant pressure compared to the capability of power systems to maintain constant voltage. Based on a typical gas network and gas turbine parameters, it takes levels of minutes for the inlet pressure of the gas turbine to drop to the protection threshold, assuming a complete interruption of the gas supply several kilometres away from the gas turbine. In contrast, it only takes levels of seconds for the gas-electric early warning system to summarize the information about the gas failure and initiate the control process of the power system. Therefore, when power generators receive the control command from the EPCC, cascading failures have not arrived at the power system, ensuring the feasibility of the gas-electric early warning system.

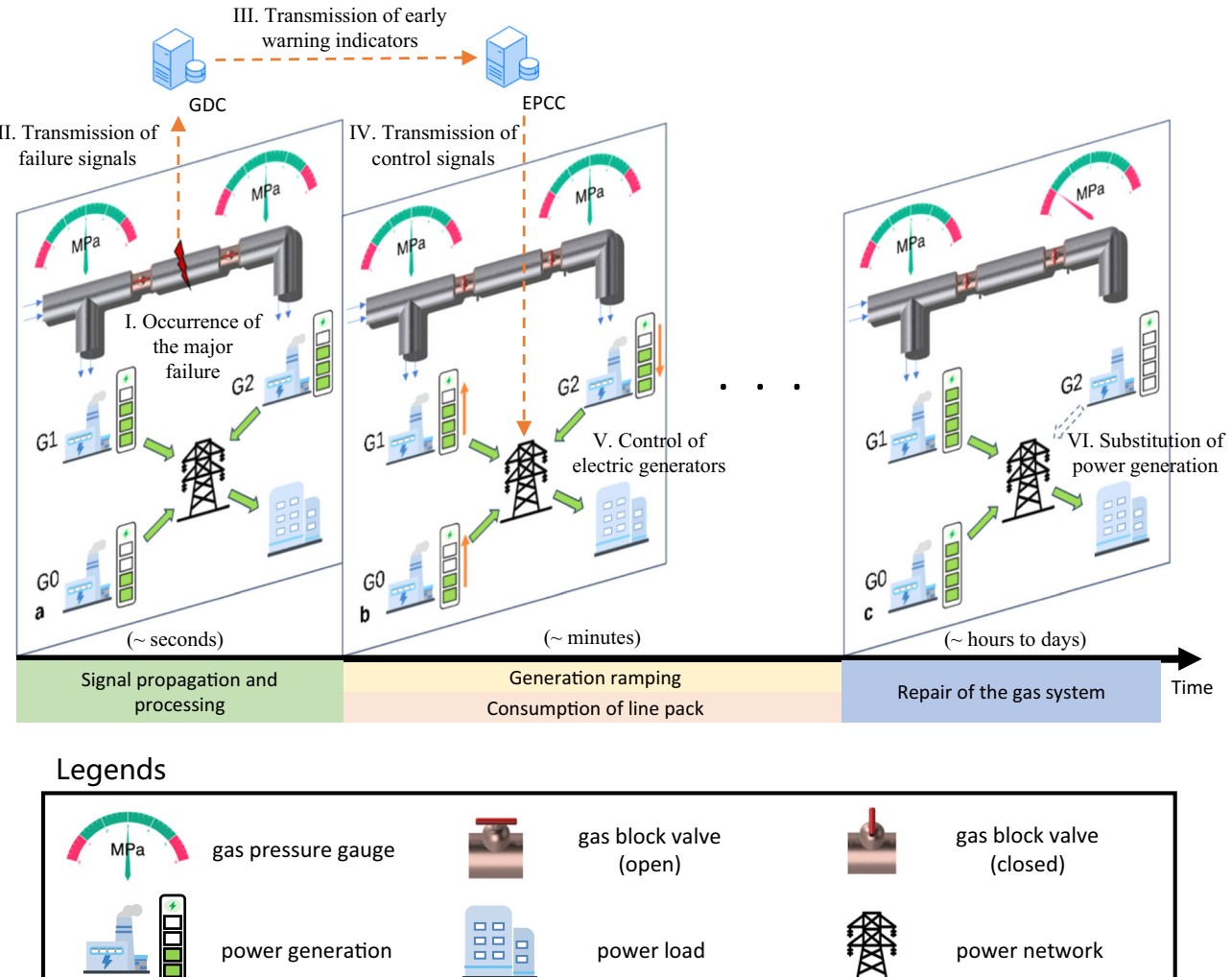

**Fig. 2 | Basic principle of the gas-electric early warning system.** When a major failure occurs in the gas network, adjacent valve chambers are closed to isolate the failure, and the gas-electric early warning system starts to work. Three power generators (G0, G1, G2) are depicted to show the power generation adjustment, among which G1 and G2 are fired by the gas network. **a** The major failure occurs in the gas network (I), and then failure signals are sent to the GDC (II). In this case, G2 is the impacted gas turbine because its connection with the gas source is completely severed. Then, early warning indicators are calculated by the GDC and sent to the EPCC (III). **b** To reduce the impact of the potential failure of G2 on the power system, the EPCC formulates the proactive control strategy of the power system based on the early warning indicators (IV). The power generation of G2 is gradually reduced while that of other generators is increased to maintain the balance of the power system (V). **c** If a complete substitution of power generation by G2 is achieved before its inlet pressure drops to the protection threshold, the security of the power supply will not be affected (VI).

Figure 2 shows the basic principle of the gas-electric early warning system. When a major failure occurs in the gas system, failure signals are sent to the gas dispatch centre (GDC). To ensure the operation of the network, the adjacent valve chambers are closed to isolate the failure, which severs the connection between certain gas turbines (defined as the impacted gas turbines) and their gas sources. Due to the gas stored in the pipeline (usually between the nearer isolation valve and the impacted gas turbine, defined as the terminal area), a period of time lapses before the impacted gas turbine is affected, while the early warning indicators issued by the GDC travel at the speed of light and soon reach the EPCC. When the EPCC starts to conduct proactive control of the power system, the inlet pressure of the gas turbine has not changed much yet. If the electric power generation of the impacted gas turbine can be fully compensated by other generators before its inlet pressure declines to the protection threshold, the initial gas network malfunction will not adversely impact the power system.

## Indicators of gas system failure

In the gas-electric early warning system, certain indicators can be defined to help the power system escape from gas system failure, which are sent from the GDC to the EPCC. Here, we introduce two early warning indicators: the *available escape time* (AET) and the *available line pack* (ALP), which provide sufficient information of the gas network malfunction for power system control. Based on these two indicators, the EPCC can formulate a suitable control strategy for the power system without other gas network information. Here, the GDC can indeed transmit comprehensive real-time information about the gas network to the EPCC, but in most countries, the management of coupled gas-electricity systems falls under two different entities, rendering this practice prone to privacy breaches.

The AET is an indicator in the time dimension, which refers to the remaining time for the power system to escape from a gas system failure; in other words, this indicator denotes the time before the inlet pressure of the impacted gas turbine drops to the protection threshold

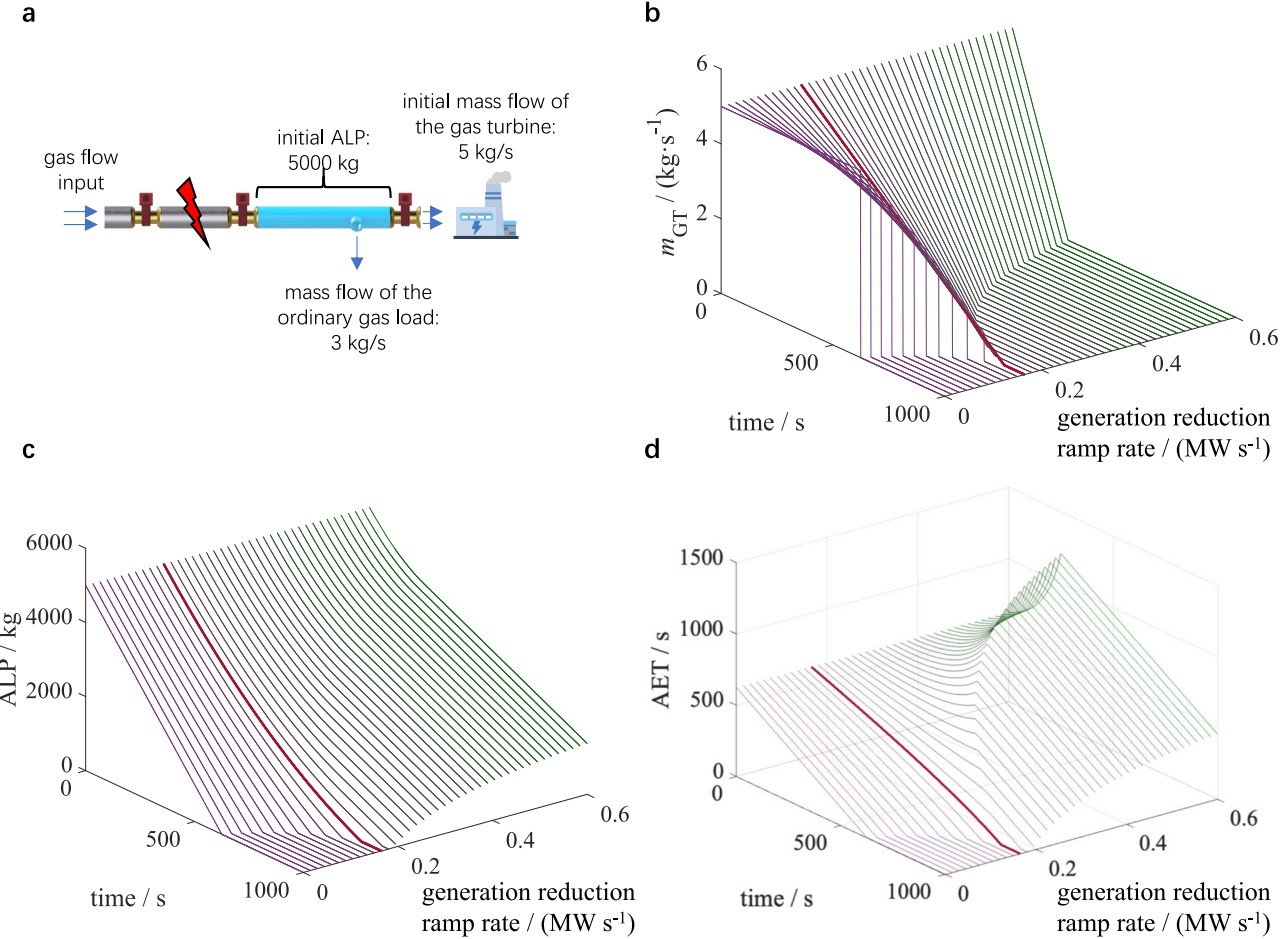

**Fig. 3 | Change in key variables after gas system failure at $t = 0$. a** Schematic diagram of gas terminal users. When the major failure occurs, both the gas turbine and the ordinary gas load can only work with the gas stored in the blue-marked pipeline. The initial ALP for the gas turbine is 5000 kg, and the initial mass flow rates of the gas turbine and the ordinary gas load are 5 kg/s and 3 kg/s, respectively. **b** Mass flow curves of the gas turbine under different generation reduction ramp rates. The curve of the critical declining ramp rate is highlighted in bold red. When the ramp rate is lower, the gas turbine is faced with a sudden shutdown due to the insufficient inlet pressure. When the ramp rate is higher, the gas turbine is actively stopped before the inlet pressure drops to the threshold. **c** ALP curves under different generation reduction ramp rates. **d** AET curves under different generation reduction ramp rates. If the ALP has not been used up when the gas turbine stops, the remaining portion will be consumed at a constant rate by the ordinary gas load. Source data are provided as a Source Data file.

according to current gas loads. Let the available escape time at time $t$ be $\text{AET}(t)$. If the generation of the gas turbine is decreasing while other gas loads remain constant, the inequation $\text{AET}(t) - \text{AET}(t + \Delta t) < \Delta t$ is satisfied, where $\Delta t$ is the time interval. This is due to the fact that the impacted gas turbine typically needs to decrease its power generation to escape from the failure, and the inlet pressure can be sustained for an extended period at a lower flow rate. If the AET is still positive when the power generation of the impacted gas turbine is controlled to zero, the gas system failure will not propagate to the power system through this turbine. To describe the emergency degree of the gas system failure itself, the initial AET (i.e. AET(0)) is defined as the *static available escape time* (SAET): the smaller the SAET, the more urgent the gas system failure is.

The location of the gas system failure relative to the impacted gas turbine is a key factor determining the SAET. In practice, if the impacted gas turbine is situated far from the failure location, it can continue operating for a long time before the inlet pressure drops to the threshold. To precisely measure this influence, we introduce the ALP as an indicator in the spatial dimension. The ALP refers to the gas line pack that remains available for the impacted gas turbine to operate without being forced to reduce the power generation after the gas system failure. As the impacted gas turbine is isolated from gas supply points, the line pack (denoted as LP) will decrease with the

consumption of gas loads. When the current line pack fails to sustain the inlet pressure within the normal range, the gas turbine will be forced to reduce its generation output or even shut down under the action of low-pressure protection device. The line pack at this time is denoted as $LP_{last}$. As an early warning indicator, the ALP can be defined as the difference between LP and $LP_{last}$. Similarly, the initial ALP measures the urgency of the gas system failure when it just occurs.

### Numerical example of the early warning indicators
To better illustrate the physical meaning of the above early warning indicators, a small gas network with one controllable gas turbine and one ordinary gas load with a fixed flow rate is constructed, where a major failure occurs in the pipeline. After the adjacent block valves are closed, the impacted gas turbine is controlled to reduce its power generation at different ramp rates until the inlet pressure drops to the protection (shutdown) threshold. Figure 3 shows curves of the AET and the ALP, as well as the mass flow rate of the impacted gas turbine. The critical point is defined as the generation reduction ramp rate of the gas turbine when its power generation is controlled to zero exactly when the ALP is exhausted. When the declining ramp rate is lower than the critical point, the ALP runs out before the power generation decreases to zero, which indicates that the impacted turbine is forced to shut down. When the declining ramp rate is higher than the critical

point, the gas turbine actively stops before the ALP is exhausted, which also limits its operation time. It can be verified that only when the power declining ramp rate matches the critical point can the gas turbine achieve its maximum operation time.

## Proactive control of the power system

The electric power system requires a real-time balance between generation and the load, which will be broken when the impacted gas turbines are forced to reduce their power generation. At this time, there exists a difference between the electric load power and the actual power generation, defined as the power deficit. Generally, the EPCC must conduct emergency control to compensate for the power deficit; otherwise, the frequency of the power system starts to drop. Common emergency control methods involve utilizing the rotating reserve capacity of generators and implementing low-frequency load shedding, and the latter is the primary reason for power outages. Modern power systems typically adhere to the N-1 principle, ensuring that the withdrawal of any single component does not result in power outages. However, as the penetration of renewables increases, the frequency inertia of the power system gradually diminishes, which means that the power deficit will lead to a faster decline in the system frequency. In such cases, even the outage of a single generator may develop into cascading failures which involve the withdrawal of other generators, eventually necessitating low-frequency load shedding by the EPCC[38,39]. Hence, when a major failure occurs in the gas network, the gas-electric early warning system must find ways to reduce the power deficit arising from the impacted gas turbines.

When information is limited to the power system, it is difficult for the EPCC to adjust the generation of the impacted gas turbines in advance, which results in the direct impact of insufficient inlet pressure of the gas turbine on the power system. In certain instances, although some gas turbines reported the situations of gas supply disruptions to the EPCC, the reports from different gas turbines were not systematic, leading to the improper utilization of rotating reserve capacities. Once the gas-electric early warning system is introduced, the EPCC can obtain an overview of the gas system failure in advance and formulate control strategies before the impact of the failure on the power system is experienced, which we refer to as *proactive control* of the power system. On the contrary, if the EPCC adjusts the output of generators only when the power deficit actually occurs, we refer to such a strategy as *passive control* of the power system. In the proactive control strategy, each normal generator with reserve capacity has a longer time to compensate for the power generation reduction of the impacted gas turbine, leading to a substantial reduction or even elimination of the power deficit, as shown in Fig. 4. In certain cases, the ALP may not be sufficient to entirely support the substitution of the impacted gas turbine by other generators. If this happens, a power deficit persists, but it is significantly reduced compared to that in the passive control strategy.

The formulation of proactive control strategies can be regarded as a special linear programming problem. The primary pursuit of proactive control is to minimize the total energy deficit during the control period. The constraints include power balance constraints, generator capacity constraints, ramp rate constraints, nonnegative ALP constraints, and maximum operating time constraints. Since the inlet pressure curve of gas turbines depends on the optimization results, the maximum operating time for each impacted gas turbine before the ALP declines to zero cannot be determined in advance. In practice, it can be set to the SAET in the first iteration—each impacted gas turbine is required to reduce its power generation to zero within its SAET. The proactive control method only containing the first iteration is called *static proactive control*. If the power deficit is fully eliminated under static proactive control, the iteration process is stopped; otherwise, the control time of the impacted gas turbines can be extended in subsequent iterations to further reduce the power deficit,

which is called *dynamic proactive control*. The extension value can be determined according to the AET of each impacted gas turbine at the specified time in the last iteration. A detailed model analysis of this part is provided in the *Methods* section.

## Case of the gas-electric early warning system in a real-world application

The application of the proposed proactive control method is demonstrated with the coupled gas–electricity system depicted in Fig. 5. Both the gas system and the electric power system are part of the actual system in Zhejiang Province. For the sake of simplicity, the coupled gas–electricity system in one city is first selected for theoretical verification, and it is assumed that the system operates in an isolated mode (i.e., the inter-city support is not considered). According to the topology of the gas network, certain major failures (such as the failures of GS and pipelines ①, ③, ④, and ⑤) will impact the electric power system (SAET < ∞), while other failures (such as the failures of pipelines ②, ⑥, and ⑦) will not (SAET → ∞). Given each major failure in the gas system, the SAET information obtained by the GDC is also shown in Fig. 5. Taking the failure of the gas source (GS) as an example, if the EPCC does not adjust the generation of any power plant, gas turbines XS and HZ will be forced to shut down 30 min 25 s and 13 min 21 s after the failure occurs in the gas system, resulting in electric power deficits of 1047.2 and 675.4 MW, respectively.

The proactive control information given by the EPCC is depicted in Fig. 6. Under the failure of GS, the power deficit is fully eliminated once the static proactive control method is adopted by the EPCC. In other words, as the SAET of each impacted gas turbine is long enough for the EPCC to substitute its power generation by other normal generators, the proactive control strategy is not iterated. Under the failure of each pipeline, the proactive control strategy can also be formulated by the EPCC based on the AET and the ALP to reduce the cascading impacts on the power system. With the help of the early warning system, the power deficit under each pipeline failure is significantly reduced. It can be seen that not all contingencies in the gas system provide a sufficient SAET for the EPCC to escape from the cascading failure. For instance, as the distance from HZ to the nearer isolation valve decreases, the SAET of HZ decreases monotonically under the failures of pipes ③, ④ and ⑤ in sequence. The calculation demonstrates that under the failure of pipes ③ and ④, the EPCC can maintain the power balance using the spare capacity within the SAET, while under the failure of pipe ⑤ it cannot—if the power generation of HZ must be reduced to zero within the SAET, the power system will lose 507.7 MW of power generation. However, since the SAET suggests a constant power generation of the gas turbine, when this power generation declines, the maximum operating time of the gas turbine will also increase. In the dynamic, proactive control scenario, if the control time of HZ is extended from 1 min 7 s to 1 min 38 s, the power deficit will be reduced maximally from 507.7 MW to 438.4 MW with the ALP exhausted. This reduction could be more substantial if the HZ was equipped with gas storage facilities to provide more line pack.

In order to verify the scalability of the gas-electric early warning system, a case study of the provincial system is also conducted as shown in Fig. 7. The provincial gas network adopts the structure of multiend gas sources, which effectively reduces the impact of a single major failure on the power system. Therefore, we simulate simultaneous failures of multiple gas sources and gas pipelines during a winter storm, as well as malfunctions of two wind farms in the power system. According to the topological analysis of the gas network, six out of twelve gas-fired generators in the power system will lose stable gas supply when the failures occur, and their SAET ranges from minutes to hours depending on the distance between the gas turbine and the failure location. We present generation curves of typical generators in the power system including gas turbines, impacted gas turbines, coal-fired steam turbines, and nuclear power plants, both in the static

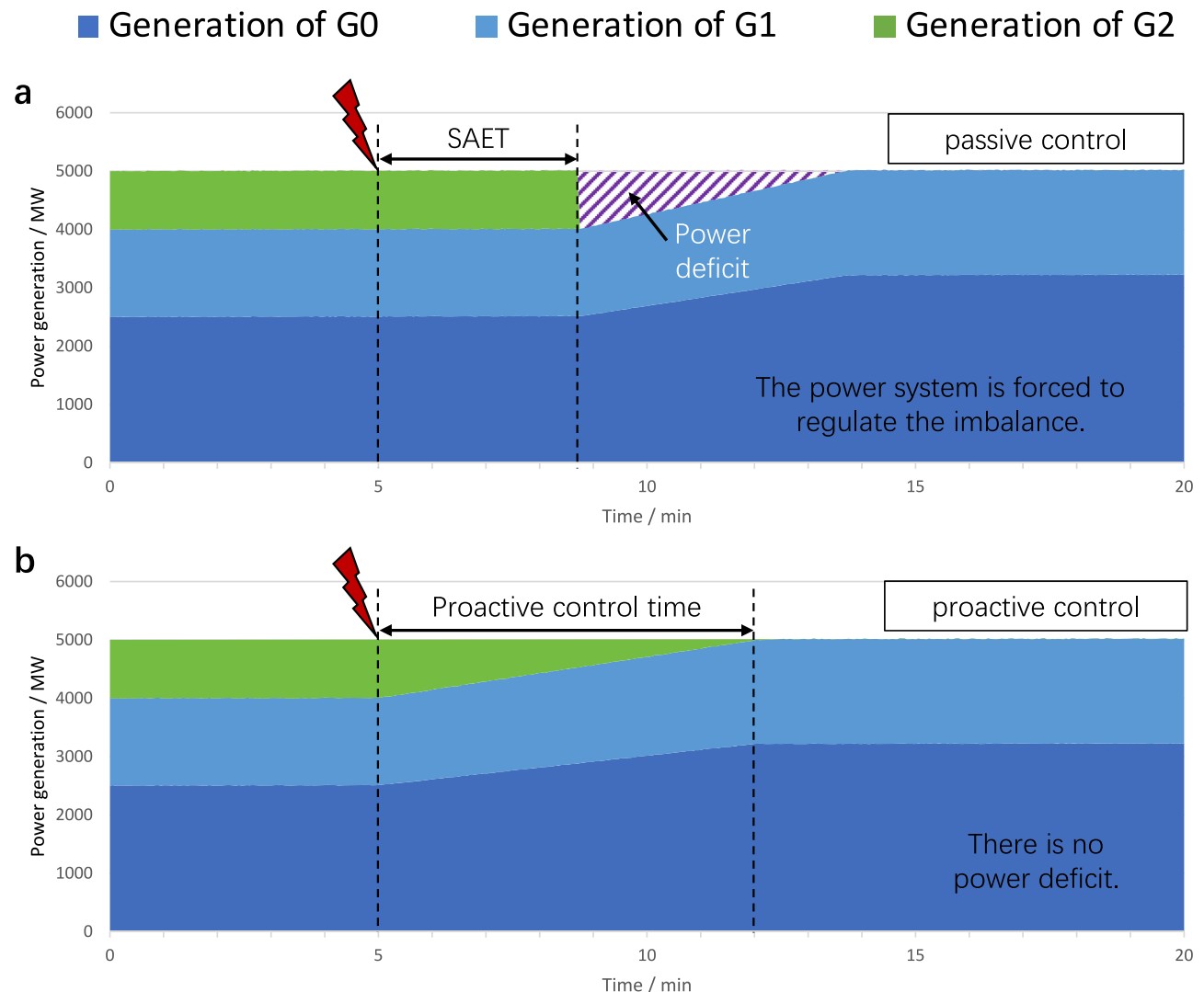

**Fig. 4 | Comparison of power control processes between passive control and proactive control.** The initial power generation of the three generators G0, G1 and G2 in Fig. 2 is selected as typical values. **a** Scenario of the passive control strategy. When the gas network failure occurs, G2 first tries keeping its original power generation for as long as possible. After the SAET, it will shut down due to insufficient inlet pressure, resulting in a power deficit. In practice, if such a power deficit is large enough, it may cause load shedding or even blackout of the power system. **b** Scenario of the proactive control strategy. G2 is required to reduce its power generation to zero before the ALP runs out, while G0 and G1 can fully offset the power deficit in real time, so there is no fluctuation in the power system. As the power generation of G2 is reduced, its maximum operation time exceeds the SAET. Source data are provided as a Source Data file.

proactive control strategy and the dynamic proactive control strategy. Among all types of normal generators, nuclear power plants are set to bear the base load, so they are not involved in the control process. Compared to the swift adjustment of gas turbines, the generation ramp rate of coal-fired steam turbines has a stricter limit due to their complex physical structures. It is demonstrated that, in contrast to passive control, which leads to a maximum power deficit of 1577.0 MW and a total energy deficit of 345.5 MWh, the maximum power deficit and the total energy deficit in the static and dynamic proactive control strategies are 1023.6 MW, 213.2 MWh and 372.0 MW, 13.8 MWh, respectively.

## Discussion

During the energy transition toward carbon neutrality, when renewable energy cannot become the main power source, gas turbines will replace many traditional coal-fired power generation sources because of their lower carbon emissions and higher control speed. In the future, when renewable energy can be applied on a large scale, owing to the development of hydrogen energy, gas pipelines and gas storage facilities will be maintained and be repurposed as important energy storage and transmission components. Due to the power system's dependence on the gas network, major failures in the gas system may lead to certain gas turbines shut down, resulting in a significant power deficit in the power system, defined as gas-electric cascading failures. In such scenarios, traditional theories of power system security, especially those based on the N-1 principle, face limitations. Real-world experiences of power outages highlight that extreme external events can disrupt fuel supply to multiple gas turbines, rendering security protection measures based on the N-1 principle ineffective. In response, we draw inspiration from the early warning principles successfully applied in the disaster response, and propose the gas-electric early warning system. This innovative system offers extended time for adjusting generator outputs, aiming to minimize the emergency control cost and reduce the likelihood of power outages.

In the gas-electric early warning system, the GDC analyses global information regarding gas major failures and notifies the EPCC in advance, so that the latter can systematically optimize the operation of

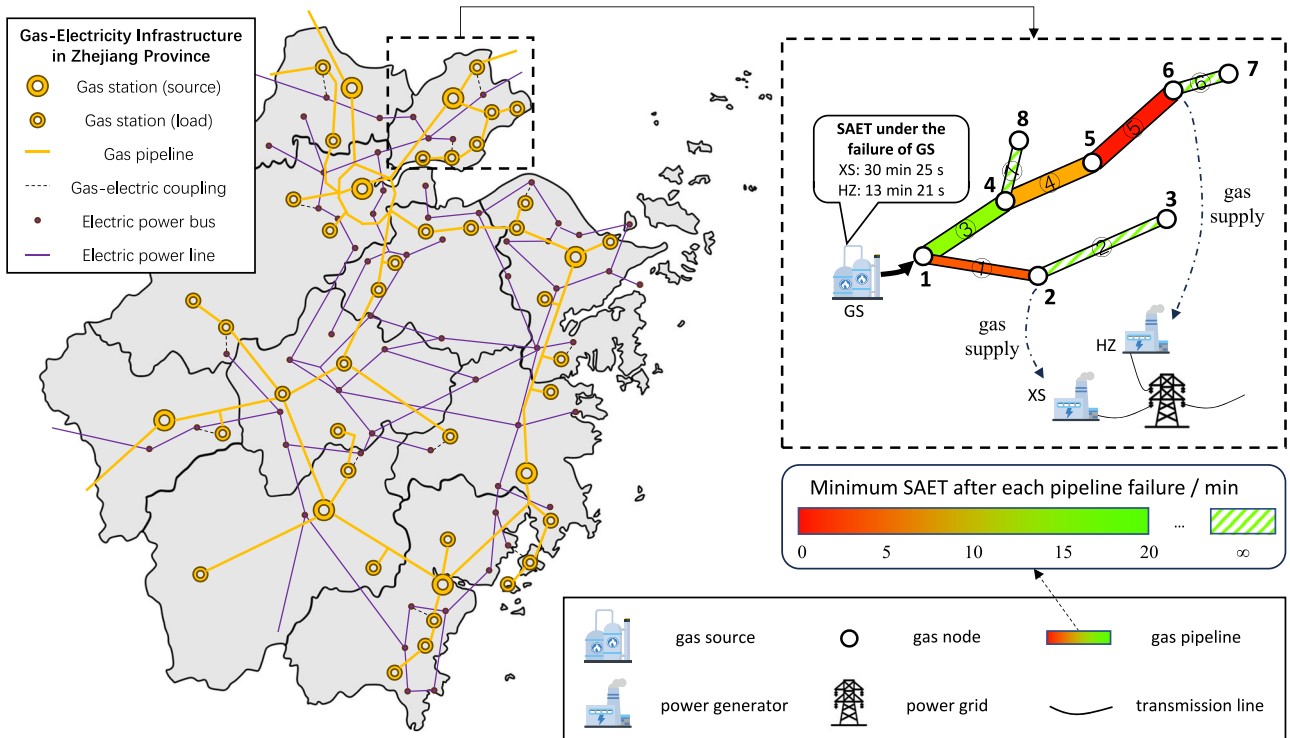

**Fig. 5 | Basics of the coupled gas–gas-electricity system and early warning of the GDC.** There are two large gas-fired power plants (HZ and XS) in the gas network, with a total power generation of 1722.6 MW. According to the topology of the gas network, the failure of the gas source (GS) or certain gas pipelines (①, ③, ④, and ⑤) will sever the fuel supply to the gas-fired power plants, and the SAETs of these failures are shown in different colours of the gas pipeline. In general, the closer the failure point is to the impacted gas turbine, the more urgent the major failure is. Source data are provided as a Source Data file.

the power system. The quantification of this global information is achieved through early warning indicators, including the AET and the ALP. In our study, these two indicators can provide sufficient information for the proactive control of the power system, eliminating the need for the EPCC to access more detailed information about the gas system. This underscores that the gas-electric early warning system does not simply combines the information of the two systems; it meets the demand of privacy protection across multiple entities. Based on the early warning indicators, the EPCC can perform proactive control instead of passive control on the power system. Compared to the passive control strategy, the proactive control strategy effectively reduces the power deficit caused by major failures in the gas network. The formulation of proactive control strategies can be modelled as a linear programming problem with unknown control times of the impacted gas turbines. By dynamically extending these control times, a larger ALP can be used to minimize the power generation deficit. In the case study of a province in East China, the power generation deficit can be reduced for different major failures in the gas network. In extreme cases, multiple failures may occur in the gas network at the same time, and the role of the gas-electric early warning method will be more significant.

The basic premise of the gas-electric early warning system is that when a major failure occurs in the gas system, the storage capacity of gas pipelines serves as a buffer for failure propagation. If this buffer is not enough to maintain the inlet pressure, the effectiveness of the early warning system will be limited. In practical engineering applications, it is possible to configure typical failure scenarios within the gas system by considering the failure probabilities of various components. This assessment serves to evaluate whether the storage capacity aligns with the requirements of the gas-electric early warning system. If inadequacies are identified, additional gas storage facilities can be installed to further ensure the feasibility of the gas-electric early

warning system, thereby enhancing the resilience of coupled gas-electric system to external extreme events.

In this paper, we mainly focus on exploring the interdependency between the power system and the gas network, and propose an innovative method to mitigate gas-electric cascading failures at the interface of the two systems. In future research, the early warning approach presented in this paper can be integrated with more detailed cascading failure models, thus yielding a more technical and comprehensive framework to address such problems. Moreover, in future energy systems, electricity, heat, gas, transportation, and other forms of energy will exhibit further coupling relationships, while their physical characteristics vary[40]. As a result, the cross-energy early warning ideas demonstrated in this paper can be extended to other energy flow combinations in future work, e.g., the coupled transportation-electricity system.

## Methods
### Hydraulic simulation of the gas network
Here, the isothermal model of gas networks is introduced to conduct the dynamic simulation, which has been shown to have negligible errors when the variation of ambient temperature is not large[41]. Due to the fast response of compressors and valves compared to pipelines, we only consider the dynamic process of gas pipelines[42]. Therefore, the core of the dynamic hydraulic simulation is to solve the initial boundary value problem of the following partial differential algebraic equation (PDAE) set:

$$
\begin{cases}
\frac{\partial \rho}{\partial t} + \frac{\partial (\rho v)}{\partial x} = 0 \\
\frac{\partial (\rho v)}{\partial t} + \frac{\partial (\rho v^2)}{\partial x} + \frac{\partial p}{\partial x} + \frac{\lambda \rho v |v|}{2D} + \rho g \sin \theta = 0 \\
p = ZRT\rho
\end{cases}
\tag{1}
$$

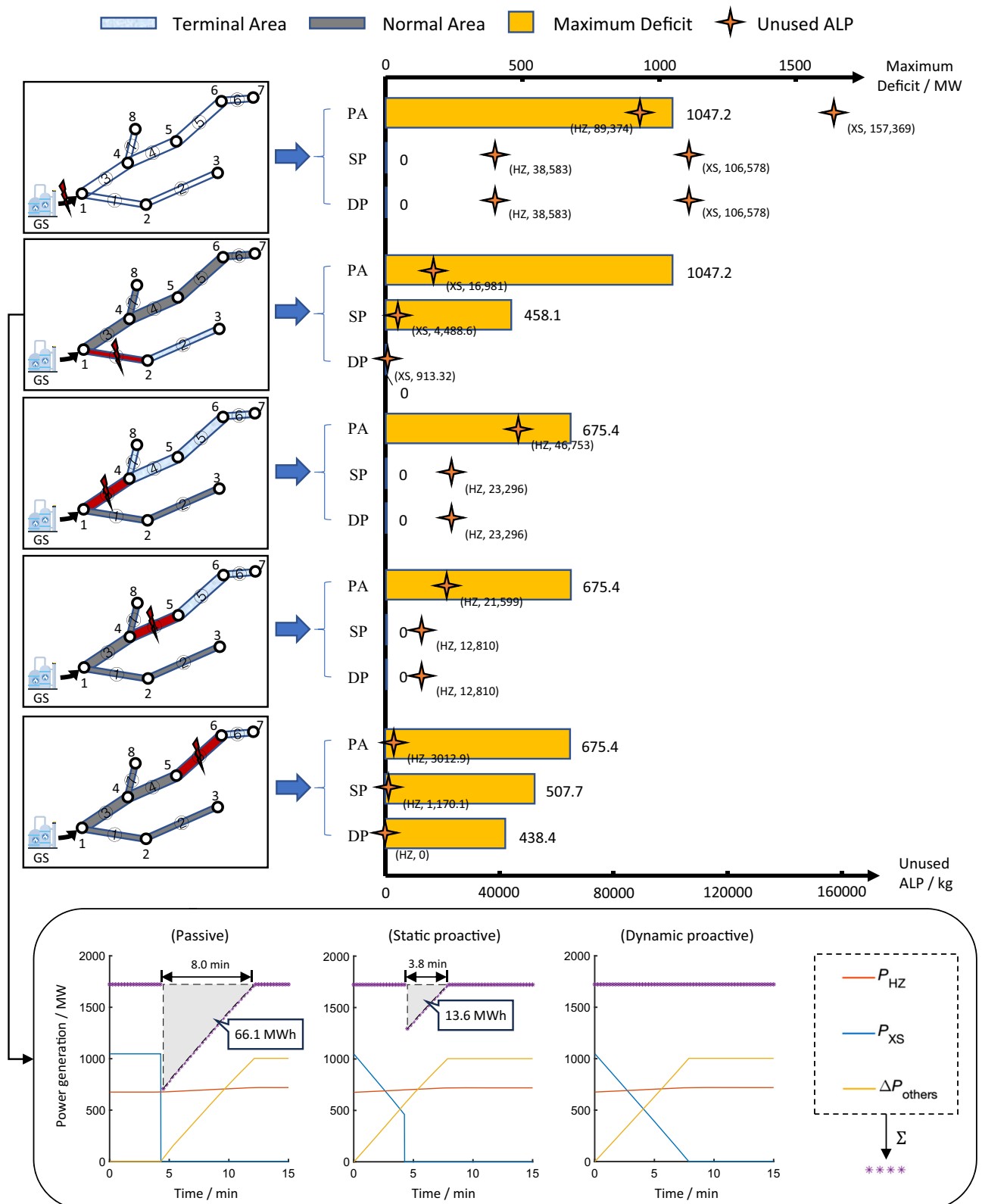

**Fig. 6 | Key results of different power control strategies of the EPCC.** This figure shows the maximum power deficit, unused ALP and typical power generation curves of XS, HZ and external power grids under different power control strategies (passive control, static proactive control and dynamic proactive control, abbreviated as PA, SP and DP, respectively). Regarding the failure of GS, the SAETs of XS and HZ are 30 min 25 s and 13 min 21 s, respectively, and a total power deficit of 1722.6 MW can be fully eliminated using the static proactive control strategy.

Regarding the failures of pipes ① and ⑤, the power deficit cannot be fully eliminated using the static proactive control strategy. When the proactive control time of XS is extended from 4 min 16 s to 7 min 53 s, the power deficit caused by the failure of pipe ① is reduced to zero, and the remaining ALP is 913.32 kg. When the proactive control time of HZ is extended from 1 min 7 s to 1 min 38 s, the ALP under the failure of pipe ⑤ is reduced to zero, and the maximum power deficit is 438.4 MW. Source data are provided as a Source Data file.

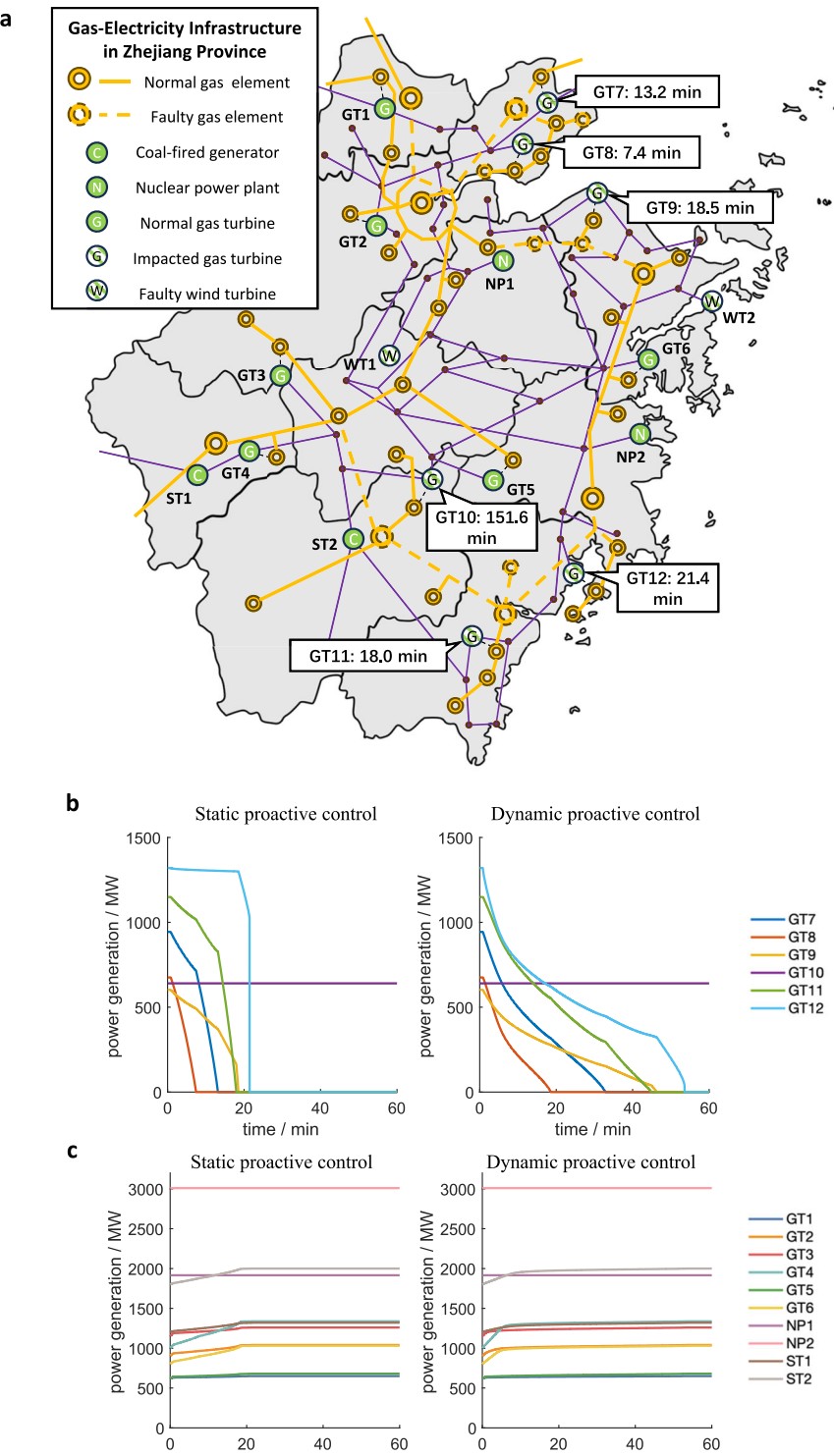

**Fig. 7 | Provincial demonstration of the gas-electric early warning system.**
**a** Malfunctions in the coupled gas-electricity system. When large-scale failures occur in the gas system, certain gas turbines are impacted and their SAET values are marked nearby. **b** Generation curves of impacted gas turbines (GT7 - GT12) in proactive control strategies. Compared to the static proactive control strategy, the dynamic proactive control strategy provides a longer power regulation time for impacted gas turbines. **c** Generation curves of normal generators (GT1 - GT6, NP1, NP2, ST1, ST2) in proactive control strategies. Typically, gas turbines have greater regulatory capabilities compared to coal-fired steam turbines, while nuclear power plants are rarely involved in the generation adjustment process. Source data are provided as a Source Data file.

The independent variables include time coordinate $t$ and space coordinate $x$, the dependent variables include density $\rho$, flow speed $v$ and pressure $p$, and the parameters include pipeline friction coefficient $\lambda$, inner diameter $D$, gravity acceleration $g$, slope angle $\theta$, compression factor $Z$, gas constant $R$ and temperature $T$. Long-distance transmission pipelines usually work in the resistance square area, where the friction coefficient $\lambda$ is only related to the pipeline parameters, not influenced by the Reynolds number. By substituting variables, we can get a simpler form of Eq. (1) similar to the telegraph equations in power transmission systems, which is essentially a wave

equation set with the propagation speed $c = \sqrt{ZRT}$[43]:

$$\begin{cases} -\frac{\partial p}{\partial x} = L_0 \frac{\partial m}{\partial t} + R_0 m \\ -\frac{\partial m}{\partial x} = C_0 \frac{\partial p}{\partial t} + G_0 p \end{cases} \quad (2)$$

The variable $m$ denotes the mass flow rate, where $m = \rho v A$. The parameters include $L_0 = 1/A$, $R_0 = \lambda|v|/(2AD)$, $C_0 = A/c^2$, and $G_0 = 0$, where $A$ is the cross-sectional area of the pipeline. Based on the finite-difference time-domain (FDTD) method[44], we suppose:

$$\begin{cases} \frac{\partial p}{\partial x} = \frac{p_{2j+1}^{[2(k+1)]} - p_{2j-1}^{[2(k+1)]}}{\Delta x} \\ \frac{\partial m}{\partial x} = \frac{m_{2(j+1)}^{[2k+1]} - m_{2j}^{[2k+1]}}{\Delta x} \end{cases} \quad (3)$$

$$\begin{cases} \frac{\partial p}{\partial t} = \frac{p_{2j+1}^{[2(k+1)]} - p_{2j+1}^{[2k]}}{\Delta t} \\ \frac{\partial m}{\partial t} = \frac{m_{2j}^{[2(k+1)+1]} - m_{2j}^{[2k+1]}}{\Delta t} \end{cases} \quad (4)$$

The variables $p_j^{[k]}$ and $m_j^{[k]}$ are the $j$-th pressure point and mass flow point at the $k$-th iterative step, respectively. Then, we obtain the following iterative difference scheme:

$$\begin{cases} p_{2j+1}^{[2(k+1)]} = \frac{C_0}{C_0 + G_0 \Delta t} p_{2j+1}^{[2k]} - \frac{\Delta t}{(C_0 + G_0 \Delta t)\Delta x}\left(m_{2(j+1)}^{[2k+1]} - m_{2j}^{[2k+1]}\right) \\ m_{2j}^{[2(k+1)+1]} = \frac{L_0 - R_0 \Delta t}{L_0} m_{2j}^{[2k+1]} - \frac{\Delta t}{L_0 \Delta x}\left(p_{2j+1}^{[2(k+1)]} - p_{2j-1}^{[2(k+1)]}\right) \end{cases} \quad (5)$$

According to Lax equivalence theorem, as long as the time step $\Delta t$ and the grid ratio $\Delta t / \Delta x$ are small enough, the above difference scheme converges. Then, we obtain the pressure and the mass flow rate at each point when the gas network failure occurs.

If the gas network has already reached a steady state, the Weymouth equation for pipelines can also be employed to conduct the hydraulic simulation. This equation implies that the change in the square of the pressure is directly proportional to the distance, providing a simple method for pressure calculations in such scenarios:

$$p_i^2 - p_j^2 = \frac{16\lambda c^2 l}{\pi^2 D^5}\left|m_{ij}\right|m_{ij} \quad (6)$$

The variables $p_i$ and $p_j$ are the pressure at the head and terminal end of the pipeline, respectively, and $m_{ij}$ is the mass flow rate from node $i$ to node $j$. The parameter $l$ is the length of the pipeline.

**Practical calculation of the early warning indicators**

To be precise, the AET and the ALP depends on the dynamic hydraulic simulation of the gas network, which gives the inlet pressure curve of the impacted gas turbine. Nevertheless, within the margin of error, we can use the average pressure of the terminal segment connected to the gas turbine as a representation of its inlet pressure when the gas turbine shuts down. This method is advantageous as the change of the former is directly proportional to the cumulative consumption of the gas load, allowing for a swift calculation of the early warning indicators which ensures a fast response of the power system. The rationality of this approximation is discussed in Supplementary Information, where numerical verifications and their results are shown in Supplementary Figs. 1 to 5 and Supplementary Tables 1 to 3.

Based on this approximation, the calculation of the initial ALP can be divided into two steps. The first step is the calculation of the initial LP, and the second step is the calculation of the final LP of each impacted gas turbine. The ALP of each gas turbine is the difference between the former and the latter. According to the gas equation of state, the gas density can be written as:

$$\rho = \frac{p}{c^2} \quad (7)$$

Hence, the initial LP (denoted as $LP_0$) can be obtained by integrating the density at each point in the terminal segment. Assuming that the average pressure of the terminal segment is equal to the protection threshold of the impacted gas turbine, the final LP (denoted as $LP_{last,i}$) can be easily obtained by multiplying the density by the volume of the gas network. Therefore, the initial ALP of gas turbine $i$ can be expressed as:

$$ALP_{0,i} = LP_0 - LP_{last,i} \quad (8)$$

If there are $n$ gas turbines in the passive gas network, and the mass flow consumed by each turbine is $m_{Gi}$ (the turbine serial numbers are sorted in ascending order of their initial ALP values), then we have the following equations:

$$SAET_1 = \frac{ALP_{0,1}}{\sum_{i=1}^{n} m_{Gi} + \sum m_d} \quad (9)$$

$$SAET_k = SAET_{k-1} + \frac{ALP_{0,k} - ALP_{0,k-1}}{\sum_{i=k}^{n} m_{Gi} + \sum m_d} \quad (10)$$

where $m_d$ is the load flow other than gas turbines in the gas network.

**Dynamic formulation of the proactive control strategy**

The objective of the optimization problem can be expressed as:

$$\min y_c = \sum_{t=1}^{T}\left(\sum_{i=1}^{N_d} c_{d,i}\left(\widetilde{P}_{d,i,t} - P_{d,i,t}\right) + \sum_{i=1}^{N_g} c_{g,i} P_{g,i,t}\right) \quad (11)$$

where $T$ is the global proactive control time, which is long enough for the power system to restore the power supply, $N_d$ and $N_g$ are the numbers of electric loads and power generators, respectively, and coefficients $c_{d,i}$ and $c_{g,i}$ are the unit adjustment costs for each load or generator $i$. Since the proactive control time is usually only at the minute level, the power generation cost is far less than the social cost caused by load shedding, so $c_{g,i}$ can be set far less than $c_{d,i}$. Variables $P_{d,i,t}$ and $P_{g,i,t}$ are the electric power consumption or generation for each load or generator $i$ at time $t$, respectively, and $\widetilde{P}_{d,i,t}$ represents the predicted electric load power without failure for load $i$ at time $t$. The above equation indicates that the objective of proactive control is to minimize the total control cost during the global control period.

For each time step $t$, the following constraints should be satisfied:

$$0 \le P_{d,i,t} \le \widetilde{P}_{d,i,t}, \forall i \in \mathbf{S}_D \quad (12)$$

$$0 \le P_{g,i,t} \le \bar{P}_{g,i,t}, \forall i \in \mathbf{S}_G \quad (13)$$

$$-R_{d,i} \le P_{g,i,t} - P_{g,i,t-1} \le R_{u,i}, \forall i \in \mathbf{S}_G \quad (14)$$

$$\mathbf{A}_g \mathbf{P}_{g,t} - \mathbf{A}_d \mathbf{P}_{d,t} = \mathbf{A}_l \mathbf{P}_{l,t} \quad (15)$$

$$\mathbf{P}_{l,t} = \mathbf{B}_l \mathbf{A}_l^\top \boldsymbol{\theta}_t \quad (16)$$

$$-\bar{P}_{l,j,t} \le P_{l,j,t} \le \bar{P}_{l,j,t}, \forall j \in \mathbf{S}_L \quad (17)$$

$$ALP_{i,T_i} \ge 0, \forall i \in \mathbf{S}_{IG} \quad (18)$$

$$P_{g,i,T_i} = 0, \forall i \in \mathbf{S}_{IG} \quad (19)$$

**Table 2 | Algorithm of dynamic proactive control method**

| | |
|---|---|
| Inputs | Early warning indicators of impacted gas turbines ($SAET_i$, $ALP_{0,i}$), parameters and variables of the power system |
| Parameters | Maximum iteration times ($iter_{max}$), control cost tolerance (tol), global control time ($T$), time-set ratio ($\alpha$) |
| Step 1 | Let the iteration number $k = 0$, and the proactive control time $T_i = SAET_i$ for each impacted gas turbine $i$. |
| Step 2 | Solve the linear programming problem (11)-(19) and get the proactive control strategy ($P_{d,i,t}$, $P_{g,i,t}$) with the control cost $y_{c,k}$. |
| Step 3 | Judge whether the control cost is acceptable or the remaining ALP for each impacted gas turbine is zero, that is, $y_{c,k} <$ tol or $ALP_{i,T_i} = 0, \forall i \in \mathbf{S}_{IG}$. If yes, go to the output step. |
| Step 4 | Let $T_{set,i} = \alpha T_i$ and calculate $AET_{i,T_{set,i}}$ of each impacted gas turbine. |
| Step 5 | Let $T_i = \max(AET_i(T_{set,i}), T_i)$. |
| Step 6 | Let $k = k + 1$. |
| Step 7 | Judge whether the iteration number $k <$ $iter_{max}$. If yes, return to step 2. |
| Outputs | Proactive control strategy ($P_{d,i,t}$, $P_{g,i,t}$) |

where:

(1) Constraints (12)(13)(14) are general element constraints, where $\mathbf{S}_D$ and $\mathbf{S}_G$ are the set of electric loads and generators, respectively. Parameters $R_{u,i}$ and $R_{d,i}$ are the maximum ramp rates of generator $i$.

(2) Constraints (15)(16)(17) are DC power flow constraints. Matrix $\mathbf{A}_l$ is the incidence matrix which reflects the connection between nodes and branches. Similarly, matrices $\mathbf{A}_g$ and $\mathbf{A}_d$ describe the location of generators and loads, respectively. For each column in $\mathbf{A}_g$ or $\mathbf{A}_d$, only the corresponding element at the node associated with the generator or the load is 1, and the other elements are 0. Vectors $\mathbf{P}_{g,t}$, $\mathbf{P}_{d,t}$, $\mathbf{P}_{l,t}$, $\boldsymbol{\theta}_t$ are all column vectors that represent the power generations, the power loads, the line flows and the node angles, respectively. Symbol $\mathbf{S}_L$ is the set of electric transmission lines.

(3) Constraints (18)(19) are gas failure constraints, where $\mathbf{S}_{IG}$ is the set of the gas turbines to be impacted by the gas-electric cascading failure. Parameter $T_i$ is the control time of each impacted gas turbine. When the control of each impacted gas turbine is completed, its power generation must be zero.

Superficially, the formulation of proactive control strategies is a linear programming problem, but the control time $T_i$ is unknown in advance, making it difficult to solve accurately. Fortunately, the target of proactive control is to obtain a good enough control strategy within the shortest time and not pursue the *best* strategy. Therefore, the following iterative method can be used to obtain an adjustment strategy suitable for engineering practice. First, determine whether the control cost is acceptable in the *static proactive control strategy* where $T_i = SAET_i$ for each impacted gas turbine. If it is, this strategy can be used for power adjustment. Otherwise, the *dynamic proactive control strategy* can further be introduced to reduce the impact of the cascading failure, where the AET of each impacted gas turbine is recalculated according to the power generation and the ALP at $T_{set,i} = \alpha T_i$, where $\alpha$ is the time-set ratio as an empirical parameter. The proactive control strategy can be reformulated by updating the proactive control time $T_i$ with the new AET at $T_{set,i}$. This approach allows for the utilization of more reserve capacity in the power system and line pack in the terminal area, further reducing the adjustment cost until it reaches an acceptable level. The algorithm of dynamic proactive control is shown in Table 2. Obviously, the calculation of dynamic proactive control strategies takes significantly longer time than the calculation of static proactive control strategies.

## Reporting summary

Further information on research design is available in the Nature Portfolio Reporting Summary linked to this article.

## Data availability

The input dataset is available for download at https://doi.org/10.5281/zenodo.11058037[45], including statistics of global gas infrastructure and parameters of the coupled gas-electricity system in Zhejiang Province. For all figures with concrete data, Source data are provided with this paper.

## Code availability

To enable replication of our work, the model and analysis code are open source, which is available at https://doi.org/10.5281/zenodo.11058037[45]. This includes the MATLAB package and all scripts used to run the analyses and create the plots in this article.

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

## Acknowledgements

This work is supported by: 1) National Key R&D Program of China with funding number 2022YFB2404000 (H. Sun and Q. Guo); 2) EPSRC/NFSC funded project: Multi-energy Control of Cyber-Physical Urban Energy Systems (MC2) with funding number EP/T021969/1 (J. Wu).

## Author contributions

H.S., Z.Q. designed the research. J.W. and Q.G. conducted the data collection and provided feedback on the scenarios and framing. F.Y. performed the analysis, created the figures and wrote the manuscript.

## Competing interests

The authors declare no competing interests.
