## [Peer Review File · Nature Communications]

REVIEWER COMMENTS

Reviewer #1 (Remarks to the Author):

In this paper authors have proposed that gas turbines will replace coal power plants for lower emissions and faster control during the energy transition so gas networks' security affects power systems. Therefore, to achieve proactive power system control, the authors introduce a gas-electric early warning method using available escape time and available line pack indicators. However, there are serious concerns about this methodology as follows:

- 1- Your claim about increased interdependency of electricity and gas networks because of transition to decarbonized gas fuels, doesn't seem true. These decarbonized fuels are not a source of energy for electric power systems. Besides, producing these fuels seems out of the scope of this paper.
- 2- Why transitioning from natural gas to decarbonized fuels should increase interdependency between gas network and electric power systems? Even if we consider these fuels as a source of energy for power plants, several transportation methods exist. Unlike natural gas which relied on pipelines, other transportation methods like trucks and ships exist for these fuels.
- 3- Literature review seems weak. Resiliency, security, reliability and other similar topics in both gas networks, power systems and integrated networks discuss these issues which are not considered here.
- 4- In the literature review please provide your results with percentage values. It is not clear that how these black outs have affected the system and why your model is necessary.
- 5- It doesn't seem that this procedure is functional in a restructured power system. Reserve capacities are also in the market and they have a price, whether purchased in the day-ahead market or real time market. So why there is a need to an entity that has control over everything and can decide on the operation of power system? Ancillary services provided by power plants and reserve markets for electricity power cannot resolve this issue? This procedure seems a step back in the restructured power markets.
- 6- In actual power systems there are different types of power plants like nuclear and hydro units to regulate the frequency. Your case study doesn't seem realistic.
- 7- You have considered a radial gas network while a ring network has higher security. Why you have considered this type of network with lower reliability? Please also provide evident of the gas network type where you proposed have faced problems, like Texas.
- 8- Time control of the system cannot be complete and realistic without considering the frequency response of power system. How do you justify this?

Reviewer #2 (Remarks to the Author):

The paper presents an important study on cascading failure problem in power-gas networks and proposes an early warning system to prevent such costly events. Although this article presents interesting scenarios and data about coupled power and gas systems and how to address the issue of coupling in case of failures; however, it has the following limitations.

1-The paper does not clearly discuss the cascade scenario and mechanism of spread of failures in both gas and power systems and instead mainly focuses on a fault in gas networks and its effects on power systems. The consequences of the initial fault in gas network and how it will be spread is only discussed through impacted gas turbines. It is not clear how the failures are assumed to propagate in power system. Also, it is not clear if the study covers cascades, which are triggered in the power system and the affect the gas system. Overall, the study focuses on the dependency of power on gas rather than the coupling effect.

2- The description of scenarios, early warning system, early warning indicators, escape strategy,... are very high level and abstract. The only place that the study gets a little deeper is in numerical examples, which still does not show the details of the approach or the theoretical significance of the study.

3-The writing and presentation of the paper needs improvement. The figure captions are long and contain so much information. Fig. 1's data seems to date back to 2020. It would be better to have it updated with more recent information. Fig. 2 is confusing and the sequence does not quite show the cascade process. Also, some sentences are not adequately clear. For example, in page 5 in sentence "...suitable escape strategy of the power system without other gas system information, which effectively protects the privacy of the GDC.", what does the privacy of GDC refer to?

Reviewer #3 (Remarks to the Author):

The paper under review presents a method for provisioning a warning system that provides the coupled electricity grid with an early warning and malfunction data when an accident occurs in the natural gas network. The purpose of the warning system is to help the electricity grid avoid or reduce electric load shedding before the gas network line-pack is exhausted to the level where gas turbine power plants have to shut down immediately. This reviewer acknowledges the authors' efforts and conveys the following observations/comments:

- The first comment is: are the authors sure that there is no such a warning system in place now? It seemed highly improbable to the reviewer that gas operators do not provide related power plants with any warnings when the gas network is unable to supply the contracted gas. Maybe the authors are encouraged to review market models with respect to this kind of warnings and underlying proactive models.
- The reviewer recommends minor changes in the title of the article so that it can represent the purpose of the study in a clearer way. The article is NOT about an early warning, as the title suggests, but it explores a method of warning of failures in the natural gas network followed by a subsequent proactive control strategy.
- Line 15: “In a gas–electricity coupled system, the identification and transmission of gas network malfunction information using communication infrastructure occur significantly faster than power system dispatch”. Why? What reference mentioned that? The sentence makes no sense.
- Line 13. “will significantly increase” is recommended instead of “significantly increases”.
- The reviewer believes that it would be helpful to explain in the introduction section that with penetration of more renewable generators, the natural gas network will have to counteract the intermittency of renewable generation, mainly through gas turbines. This duty for gas networks was not significant in the past. The new duty will make the gas networks more exposed to fluctuations, so a higher possibility of faults/failures can be expected. That is why gas networks will need much more accurate and sophisticated control programs.
- Lines 66 and 67. The reviewer expects a literature review on the security of coupled networks. However, only two references are mentioned insufficiently.
- Line 52, Figure 1: “In this decade, although a general consensus has been reached on reducing global carbon emissions, ...”. Please note that in the reviewer’s opinion moving toward generating electricity from natural gas does not necessarily contradict the consensus on reducing greenhouse gases since natural gas-fired power plants produce less amount of greenhouse gases and other pollutants (with respect to other fossil-based power plants) in addition to paving the way for penetration of more renewable electricity, as it is also pointed out in the article. Maybe the authors want to correct the sentence by omitting the part that was mentioned in this comment.
- Line 58. The authors are encouraged to provide a stronger reasoning to why power blackouts may become more probable in the future. They are recommended to mention that with the growth of hydrogen energy systems and the large-scale water electrolyzers coming into operation, the mutual energy exchange between gas networks and electricity grids will increase significantly, so a failure in one can have a severer impact on the other....
- Table one and also in the text: The reviewer recommends not assessing a power loss only based on the maximum megawatts. It would provide a much better understanding if the authors also provide the mean duration of power loss and the total electric energy loss (megawatt hours not supplied).
- Line 127. Please explain why this inequality applies and what it means exactly.

- Line 138. Maybe it is better to write “as the gas turbine is isolated from gas supply points, the line-pack will decrease....”.
- The reviewer suggests adding “which can be regarded as the minimum escape time” after “in the first iteration”.
- Line 183: The sentence has to be discussed in greater detail based on the Table 3 algorithm. The reviewer was unable to fully understand the algorithm because of the lack of explanation on parameters Tset and time-set ratio (alpha). Please clarify by adding new sentences.
- Line 195: It is highly recommended to plot the diagram of the impacted power plant control time versus the ramp rate, showing the maximum operation time and the critical ramp rate. Furthermore, it is suggested to mention the capacity of the gas turbine under discussion and to omit the star sign from the labels in Figure 4 (MW s⁻¹ and kg s⁻¹ are enough).
- Figure 4.d. The reviewer was unable to understand the descending behavior of the curves after the bold red curve (i.e. ramp rates above the critical ramp rate). It needs clarification.
- Figure 4. It seems that “generation reduction ramp rate” is a more accurate label than “ramp rate”.
- Line 208 and similar cases. Please replace “gas-electricity coupling system” with “coupled gas-electricity system”.
- Figure 5. The label of pipelines is hard to read. Using alphabetic characters is suggested.
- Figure 6. Please mention the unit for unused ALP. In addition, it is highly recommended to plot power generation diagrams for the second case as well (failure of line 1). Also, the authors may want to report the duration of power deficit and the total load shed as comparison parameters in a table for all of the 5 cases.
- Line 222. Please add the word “after” after the word “shut down”.
- Last case in Figure 6. Please explain why such a short control time is adopted for the power plant HZ (00:01:38 and 00:01:07) while in Line 223 it was mentioned that HZ has a SAET of 00:13:21 (13 minutes and 21 seconds)
- Line 226. How is it proved that EPCC had enough time and available reserve capacity to substitute the lost generations?
- Line 227. According to the provided algorithm in the Methods section, the word necessarily is not required here.
- Line 292. The reviewer believes “unknown” is more accurate than “uncertain”.
- Line 310. References are required.
- Line 314. Please provide references for equations (1).
- Line 331. Please support this hypothesis with underlying gas dynamic equations.
- Line 406, The sentence beginning with “otherwise” needs more explanations and discussion.

- Table 3. Please discuss the time-set ratio.
- A medium size case study has the ability to better show the effectiveness of the method discussed in this paper.
- For the fastest expansion of knowledge, the reviewer suggests making the data related to case studies available to other researchers.
- All the parameters and variables that are used in equations must be introduced.
- The equations related to gas pressure are not valid in general. So for all equations, authors must discuss governing assumptions.
- Fig 3: the passive control: is not like illustrated in the figure, because the gas pressure sensors at the power plants can capture the pressure drop immediately at the time of gas pipe failure. You may be interested to know that they can even estimate the location of pipeline failure with an acceptable precision.
- Line 212: as the gas runs through a network of pipes, not a single pipe, it is possible that the pressure drop due to pipeline failure might be compensated by other lines connected to each other. Therefore the one line assumption is too simplistic in a way that change the nature of the event.
- Line 81: the frequency of events (gas network failure) is 4 times in 9 years, all over the world and it has not been repeated for any of locations. It shows that local EPCCs might figure out a way to solve the problem or basically it is not an important issue to address.
- Figure1-a,b: bad illustration. Hard to understand at the first sight and it is also hard to interpret after even reading the caption. In fact, it is not following the best practice of data visualization.

Reviewer #4 (Remarks to the Author):

This paper develops a method to better deal with the failures of the gas network that can propagate to the power system and cause outages and blackouts. The paper computes two metrics: available escape time (AET) and available line pack (ALP) that are communicated to the power system control center, as soon as a failure in the gas system occurs. These metrics are integrated within an optimization model that produces proactive control strategies, to reduce the negative impacts on the power system.

The increasing interdependence between the gas and power systems is an important problem, adding to the complexity of reliable operation of the two systems. However, it is not clear if the method proposed in the paper is sufficient for addressing this challenging problem. I suggest that authors add a subsection to clearly describe and discuss the particular applications of their proposed method. For instance, clearly

state the type of events, where the method is useful and the events where the method would not be so useful. Additionally, a discussion of the strengths and weaknesses/shortcomings seems to help the readers better understand the scope of the work and its applications.

Other than my above comment on the contributions and scope of the work, I provide a few additional comments below:

1. Consider adding a discussion about N-1 reliability standards and their shortcomings relative to the work presented in this paper.

2. Elaborate why and if AET and ALP provide sufficient information for power system operation, regarding the failures in the gas network. If not, what sort of additional information may be needed?

3. The authors seem to have ignored a considerable body of literature that aims to solve a similar problem. At least a brief literature review, comparing this paper to the existing literature is required to clarify the contribution of the work. Look, for instance, at Los Alamos National Laboratory's work on power and gas networks.

4. Some of the events listed in Table 1 were so large and widespread that I doubt could be alleviated by the proposed method. Additionally, in some cases a failure of the grid could lead to the failure of electric powered gas pumps, which has happened in the past in New Mexico, causing further issues in the power grid. Please elaborate on how the proposed method would or would not help in historical events, such as the ones presented in Table 1.

Response to the Reviewers

Response to the Reviewers: Reviewer 1

Comments to the Author:

- *In this paper authors have proposed that gas turbines will replace coal power plants for lower emissions and faster control during the energy transition so gas networks' security affects power systems. Therefore, to achieve proactive power system control, the authors introduce a gas-electric early warning method using available escape time and available line pack indicators. However, there are serious concerns about this methodology as follows.*

Response:

We sincerely thank you for your careful review and valuable comments. All of your comments are responded to point by point below and the original manuscript is carefully revised according to your comment. We have revised the key concepts and reorganized the simulation results to make the merit and novelty of the work clearer. We hope that we have adequately addressed your concerns.

Main Comments (#1):

- *1. Your claim about increased interdependency of electricity and gas networks because of transition to decarbonized gas fuels, doesn't seem true. These decarbonized fuels are not a source of energy for electric power systems. Besides, producing these fuels seems out of the scope of this paper.*

Response:

Thank you for your valuable suggestion.

We acknowledge the identified issue and have made the necessary adjustments. The phrase in the abstract has been revised to "the transition from coal and oil to gas fuels" to avoid ambiguity. The gas fuels here include both natural gas and decarbonized gas fuels (e.g., hydrogen). We know that because fossil fuels cannot be completely removed at present, the transition from coal and oil to natural gas can reduce the carbon emission intensity of the energy system, which is what many developing countries (e.g., China) are working towards.

In the future, gas storage will be a way of power system energy storage technology: in the valley of power consumption, excess renewable energy generation is converted into decarbonized gas storage in the pipeline, while in the peak of power consumption, the decarbonized gas is converted back into electricity by gas turbines. This is why we refer to decarbonized gas fuels as energy storage and transmission components. In order to achieve this goal, some research on decarbonized gas fuels has focused on incorporating alternative gas into natural gas infrastructure, where existing gas pipelines can still be utilized to reduce investment costs (e.g., ref 1.1~1.2). It is important to note that in this paper, we highlight the interconnection between the power grid and the gas pipeline system. Whether the fuel is natural gas or decarbonized gas fuels, as long as it is utilized in gas turbines, it is applicable to the scenarios described in our manuscript.

After modification, the abstract is now as follows:

There is growing consensus that gas-fired power generation will play a crucial role during the transition to net-zero energy systems, both as an alternative to coal- or petroleum-based power generation and as a flexibility service
--

provider for power systems. However, malfunctions of gas networks have caused several large-scale power blackouts in the past. The transition from coal and oil to gas fuels significantly increases the interdependence between gas networks and electric power systems, thus raising the risks of more frequent and widespread power blackouts due to the malfunction of gas networks. In a coupled gas–electricity system, the identification and transmission of gas network malfunction information, followed by the redispatch of electric power generation, occur notably faster than the propagation and escalation of the malfunction itself, e.g., significantly diminished pressure. On this basis, we propose a gas-electric early warning system that can reduce the negative impacts of gas network malfunctions on the power system. An optimal proactive control strategy of the power system is also formulated based on the early warning indicators. The effectiveness of this method is demonstrated via case studies of a real coupled gas–electricity system in East China.

Reference:

1.1 Abeysekera, Muditha, et al. "Steady state analysis of gas networks with distributed injection of alternative gas." *Applied Energy* 164 (2016): 991-1002.

1.2 UK Research and Innovation. Ofgem Strategic Innovation Fund announces 10 funded projects (2023). <https://www.ukri.org/news/ofgem-strategic-innovation-fund-announces-10-funded-projects>

Main Comments (#2):

- 2. *Why transitioning from natural gas to decarbonized fuels should increase interdependency between gas network and electric power systems? Even if we consider these fuels as a source of energy for power plants, several transportation methods exist. Unlike natural gas which relied on pipelines, other transportation methods like trucks and ships exist for these fuels.*

Response:

Thank you for your valuable suggestion.

In our previous response, we have answered the reasons for the increasing interdependency between power systems and gas systems. The development of gas systems includes both the increased extraction of natural gas and the adaptation of decarbonized gas fuels to natural gas infrastructures.

In terms of cost, pipeline transportation stands out as the most economically efficient method for fluid fuel transport. This preference is evident in the widespread use of oil pipelines and natural gas pipelines worldwide. Indeed, decarbonized gas fuels can also be transported by trucks and ships (e.g., ref 2.1), these modes of transportation fall outside the scope of our research. As far as we know, many coastal provinces in China import a lot of liquefied natural gas (LNG) by ship and truck, but they will eventually be transported to terminal users by pipeline. In the UK, Project Union is trying to connect hydrogen production, storage and demand to enable net zero and empower a UK hydrogen economy, where the decarbonized gas fuel is transmitted by pipeline (ref 2.2). In a word, our study primarily concentrates on the coupling between electric power systems and gas pipeline systems, and the development of gas pipeline infrastructure has been illustrated with data in our article.

In addition, we have added examples of hydrogen technology by pipeline in the introduction section of the manuscript, as shown below:

For example, in early 2021, Shell, Mitsubishi Heavy Industries, Vattenwall and Wärme Hambruge signed a letter of intent to develop renewable hydrogen production in Hamburg, where the existing gas network will be expanded to accommodate the transport of hydrogen^{Error! Reference source not found.}. In July 2023, UK Research and Innovation (UKRI) and Office of Gas and Electricity Markets declared a funding allocation of £95.3 million to push forward

10 large-scale demonstration projects, many of which aim to enhance the automation of gas systems and investigate the feasibility of integrating green hydrogen into the existing gas networks^{Error! Reference source not found.}.

Reference:

- 2.1 Hydrogen gas turbine offers promise of clean electricity. <https://www.nature.com/articles/d42473-022-00211-0>
2.2 ProjectUnion - National Gas. <https://www.nationalgas.com/document/139641/download>

Main Comments (#3):

- 3. Literature review seems weak. Resiliency, security, reliability and other similar topics in both gas networks, power systems and integrated networks discuss these issues which are not considered here.

Response:

Thank you for your suggestions.

In the initial version, we considered the matter of public interest and opted not to delve extensively into existing research on energy system security or resilience. However, inspired by your insights, we realized that the omission of this section was diminishing the prominence of our research. Now we have integrated the relevant content based on your suggestion.

We have added the following paragraph in the Introduction part:

So far, some research institutions, such as LANL, NREL and EMPA, have put forward theories related to energy security or resilience from different angles to assess or enhance the capacity of energy systems to sustain regular energy supply amidst disturbances. Grouped by disturbance sources, these theories can be classified as dealing with internal disruptions (e.g., gas leakage^{Error! Reference source not found.}) or external disruptions (e.g., extreme weather events^{Error! Reference source not found.}, ^{Error! Reference source not found.}, cyber attacks^{Error! Reference source not found.}). Organized chronologically, they focus on the ability of energy systems to prepare for, absorb, adapt to, and recover from disruptive events before and after the disturbance^{Error! Reference source not found.}, ^{Error! Reference source not found.}. In addition to traditional research on single energy resources, the exploration on the comprehensive resilience of coupled energy systems has also made breakthroughs (e.g., hurricane analysis^{Error! Reference source not found.}, disaster avoidance^{Error! Reference source not found.}).

Main Comments (#4):

- 4. In the literature review please provide your results with percentage values. It is not clear that how these black outs have affected the system and why your model is necessary.

Response:

Thank you for your valuable comments.

To sustain public interest in our paper, we have provided a concise overview of "security" and "resilience" related to energy systems. These encompass studies spanning pre-, during-, and post-disturbance phases across diverse energy systems, making it challenging to quantify their contribution with simple percentage values. In addition, many studies do not provide simple percentage values on safety improvements, which may be detrimental to our citation if we want to compare these values.

In response to your valuable suggestion, we have emphasized the salient features of our work and underscored its advantages over conventional research. Notably, we highlight the application of the early warning principles, which is the key difference between our study and other conventional studies. These principles have proven effective in other domains but have not been applied to the security of energy systems yet. Therefore, we provide a demonstration of the gas-electric early warning system which

significantly reduces the impact of the gas system failure on the power system:

However, most of the above studies play a limited role in the prevention of gas-electric cascading failures for the lack of real-time coordination between the two energy entities when the gas network malfunction occurs. This is related to the temporal misalignment of the failure concerning the two energy systems: the failure has already transpired in the gas system, while the power system is still on the verge of the failure. Current theories about energy resilience are less prone to take the advantage of this propagation delay in the medium, which has actually been demonstrated by the earthquake early warning system in the area of geology and disaster response^{Error! Reference source not found., Error! Reference source not found.}

Main Comments (#5):

- 5. *It doesn't seem that this procedure is functional in a restructured power system. Reserve capacities are also in the market and they have a price, whether purchased in the day-ahead market or real time market. So why there is a need to an entity that has control over everything and can decide on the operation of power system? Ancillary services provided by power plants and reserve markets for electricity power cannot resolve this issue?*

Response:

Thank you for your valuable comments.

In regions with imperfect power markets (e.g., China), centralized control by the EPCC remains dominant, limiting the market's ability to respond effectively to large-scale failures. Even in restructured power systems with developed power markets, like the United States, gas-electric cascading failures have led to widespread blackouts. For instance, during the Texas winter storm of 2021, numerous gas turbines experienced unplanned outages due to disruptions in gas supply. This resulted in a significant electricity supply shortage, causing electricity prices to surge dramatically. During such periods, the power market struggles to fulfill its normal regulatory role, leading to many contract failures in completing delivery. In such scenarios, it is beneficial to have an EPCC that can decide on the operation of the entire power system to reduce load cutting.

Moreover, even if the reserve capacity market continues to play the role in such scenarios, our proposed approach can mitigate the need for reserve capacity, especially rapid-controlled reserve capacity, in the power system, thereby reducing the cost associated with reserve capacity. Illustrated in Figure 4 in the manuscript is the comparison between passive and proactive control. In passive control, the standby unit must ascend the slope at its maximum speed to minimize the impact of the power deficit on system frequency, demanding a high adjustment rate. Conversely, in proactive control, the power system calls these standby units for a longer time, so there is no need to prepare much rapid-controlled reserve capacity for the gas-electric cascading failure.

In order to better show that the procedure presented here is still functional in restructured power systems, we have modified the factual arguments in the introduction as follows:

In fact, similar incidents took place in the United States in 1983, 1989, 2003, 2004, 2006, 2008, 2009, 2010, and 2011 due to adverse weather conditions^{Error! Reference source not found.}. Additional factors such as pipeline leakage and illegal operations may also lead to gas-electric cascading failures, as exemplified in California (2015)^{Error! Reference source not found.} and Taiwan (2017)^{Error! Reference source not found.}. In short, power outages caused by gas network malfunctions are presently widespread and are anticipated to pose an increasing risk in the near future.

Figure 4 Comparison of power control processes between passive control and proactive control.

Main Comments (#6):

- 6. In actual power systems there are different types of power plants like nuclear and hydro units to regulate the frequency. Your case study doesn't seem realistic.

Response:

Thank you for your valuable suggestion.

We have enhanced the realism of our current example system by incorporating a comprehensive description of the entire network structure. Furthermore, we have included a large-scale numerical example featuring a province-wide network, focusing on the study of typical generation control curves of different types of generators such as normal gas turbines, impacted gas turbines, coal-fired steam turbines, and nuclear power plants. It's essential to note that, in adherence to the confidentiality requirements of the power company, we have selectively disclosed critical information that significantly influences the control outcomes.

The supplementary content about the large-scale numerical example is as follows:

In order to verify the scalability of the gas-electric early warning system for large-scale systems, a case study of the provincial coupled gas-electricity system is also analyzed (shown in Figure 7). The provincial gas network adopts the structure of multiple gas sources, so it is difficult for a single major failure of gas sources or pipelines to have a substantial impact on gas supply. Therefore, we simulate simultaneous failures of multiple gas sources/pipelines during a winter storm. We present power generation curves for typical generators in the static proactive control strategy, including normal gas turbines, impacted gas turbines, coal-fired steam turbines, and nuclear power plants. In this strategy, the impacted gas turbines are required to control their power generation gradually to zero within their respective SAET, while other normal generators compensate for the power shortfall. Among these normal generators, nuclear power plants are set to bear the base load, so they are not involved in the power control process. Compared to swift adjustment of gas turbines, power generation curves of coal-fired steam turbines exhibit greater stability due to the constraint of their complex physical structures. It is demonstrated that, in contrast to passive control, the energy deficit of the provincial system under this strategy is reduced from 512.0

MWh to 270.2 MWh within 2 hours after the failure.

Fig. 7 | Provincial demonstration of the gas-electric early warning system. This figure shows power generation curves of different types of generators given multiple gas major failures in an external extreme event, including normal gas turbines (GT1, GT2), impacted gas turbines (GT3, GT4), coal-fired steam turbines (ST1, ST2) and nuclear power plants (NP1, NP2). Generally, gas turbines are more capable of regulating than coal-fired steam turbines, and nuclear power plants are usually not involved in regulating. Typically, gas turbines exhibit greater regulatory capabilities compared to coal-fired steam turbines, while nuclear power plants are rarely involved in the generation adjustment process.

Main Comments (#7):

- 7. You have considered a radial gas network while a ring network has higher security. Why you have considered this type of network with lower reliability? Please also provide evident of the gas network type where you proposed have faced problems, like Texas.

Response:

Thank you for your valuable comments.

Following your suggestion, we have enriched our example by incorporating the actual natural gas system in Zhejiang Province. This system comprises ring networks and eight gas sources, ensuring high reliability and bringing our study closer to the real-world scenarios of large-scale power outages mentioned in the introduction, as shown in Figure 5 and Figure 7 in the manuscript. Consequently, our example now encompasses both a small isolated network under N-1 failure and a large interconnected network under N-m failure, contributing to the enhanced credibility of the numerical examples.

The change in Figure 5 is shown below:

Main Comments (#8):

- 8. Time control of the system cannot be complete and realistic without considering the frequency response of power system. How do you justify this?

Response:

Thank you for your valuable suggestion.

After careful consideration, we have opted not to introduce the concept of frequency control in our paper. Our concern stems from the potential impact on public interest, as well as the recognition that the inclusion of frequency control would lead to variations in the model construction based on the frequency regulating elements within the system. Instead, we have chosen to refer to the widely used "power deficit" index in resilience assessments of energy systems. This index is commonplace in the evaluation and enhancement of energy system resilience, serving as a valuable metric to illustrate the contribution of the proposed method in this paper to the resilient enhancement of coupled gas-electricity energy systems.

In other words, our gas-electric early warning system is aimed at alleviating the impact of a gas system failure on the power system, rather than delving into the power system's response to such an impact—the latter is a topic well-explored in numerous professional studies on power system frequency control. We firmly believe that the concept of power deficit serves as a comprehensive measure of the magnitude of this impact. Additionally, it effectively characterizes the subsequent regulatory measures taken by the power system to mitigate this impact, which is sufficient for our study.

In order to better show the impact of the gas network failure on the power system, we added the definition of the power deficit:

At this time, the EPCC will rapidly increase the output of other generators to compensate for the power deficit (defined as the difference between the required power load and the actual power generation); otherwise, the power system may enter a state of emergency (i.e., the frequency drops below the safety threshold).

At the end of the response, we thank you again for offering us so many valuable suggestions that help us improve our work. We sincerely hope that we have adequately addressed your concerns.

Response to the Reviewers: Reviewer 2

Comments to the Author:

- *The paper presents an important study on cascading failure problem in power-gas networks and proposes an early warning system to prevent such costly events. Although this article presents interesting scenarios and data about coupled power and gas systems and how to address the issue of coupling in case of failures; however, it has the following limitations.*

Response:

We sincerely thank you for your careful review and the valuable suggestions to improve the quality of the paper! According to your suggestions, we have made corresponding changes to the manuscript. The following are the answers to your questions and concerns. We hope that we have adequately addressed your concerns.

Main Comments (#1):

- *1. The paper does not clearly discuss the cascade scenario and mechanism of spread of failures in both gas and power systems and instead mainly focuses on a fault in gas networks and its effects on power systems. The consequences of the initial fault in gas network and how it will be spread is only discussed through impacted gas turbines. It is not clear how the failures are assumed to propagate in power system. Also, it is not clear if the study covers cascades, which are triggered in the power system and the affect the gas system. Overall, the study focuses on the dependency of power on gas rather than the coupling effect.*

Response:

Thank you for your valuable comments.

We intentionally do not delve into the specific paths of failure propagation within the gas system or the power system but focus more on the blocking process of cascading failures. This is mainly because comprehensive coverage of failure propagation paths in single energy systems can be extensively found in specialized literature on gas transmission systems and electric power systems.

This paper mainly focuses on the one-way propagation of major failures from the gas system to the power system, with a primary emphasis on showcasing the "early warning system". This choice stems from the contrasting speeds of dynamics in different systems—the gas system exhibits a slow failure propagation speed, while the power system operates at a faster control speed. This distinction forms the foundational basis of the early warning system, which is also the core innovation of our article.

The term "coupled" in our paper primarily refers to the conventional terminology of a coupled gas-electric system, rather than emphasizing the bidirectional interaction between the two system. Indeed, in the 2021 winter storm in Texas, the power supply loss led to the shutdown of some natural gas infrastructure, thereby exacerbating the scale of the power outage. Nevertheless, it is crucial to note that the underlying premise in this scenario is still the failure of the gas system, which triggers the shutdown of the gas turbine, ultimately impacting the power supply. The blocking of gas-electric cascading failures is the fundamental way to mitigate such power blackouts.

Furthermore, your suggestion has sparked a valuable idea for our future research endeavours. As P2G (Power-to-Gas) technology advances, the dependence of the gas system on the power system will also grow. This inspires us to explore bidirectional fault-blocking methods in future investigations.

In order to better emphasize the theme of this article, we have reorganized the factual and rational arguments used in the introduction that power systems are more likely to be impacted by gas network failures, as follows:

However, compared to carbon emission reduction, the security of energy systems has not attracted enough attention from the public. With the expansion of gas networks, gas system malfunctions are now more likely to evolve into electric power outages. In February 2021, large-scale blackouts occurred in Texas and the south-central United States due to extremely cold weather^{Error! Reference source not found.}. Between February 8th and 20th, a gas generating capacity of 106,585 MW encountered unplanned outages or derates, primarily stemming from issues in the gas system^{Error! Reference source not found.}. **Despite there were advance notifications to each impacted gas turbine, the power system had to resort to load shedding in response to the crisis.** In the post-investigation report, the gas-electric interdependency was considered one of the most significant factors leading to the accident. **In fact, similar incidents took place in the United States in 1983, 1989, 2003, 2004, 2006, 2008, 2009, 2010, and 2011 due to adverse weather conditions^{Error! Reference source not found.}. Additional factors such as pipeline leakage and illegal operations may also lead to gas-electric cascading failures, as exemplified in California (2015)^{Error! Reference source not found.} and Taiwan (2017)^{Error! Reference source not found.}. In short, power outages caused by gas network malfunctions are presently widespread and are anticipated to pose an increasing risk in the near future.** Firstly, the transition from coal and oil to natural gas increases the dependence of electric power systems on gas networks, **and even the latter is gradually replaced by decarbonized gas fuels, e.g., blue or green hydrogen or ammonia, gas networks persist^{Error! Reference source not found., Error! Reference source not found.}**. Secondly, an increasing number of studies indicate that extreme weather events have become more frequent than before^{Error! Reference source not found., Error! Reference source not found.}, which may not only directly do harm to power systems but also threaten them by affecting the transportation of gas fuel^{Error! Reference source not found.}. Thirdly, global geopolitical tension amplifies the likelihood of human factors impacting the security of gas supplies, such as the Nord Stream gas pipeline explosion^{Error! Reference source not found.}.

Main Comments (#2):

- 2. *The description of scenarios, early warning system, early warning indicators, escape strategy, ... are very high level and abstract. The only place that the study gets a little deeper is in numerical examples, which still does not show the details of the approach or the theoretical significance of the study.*

Response:

Thank you for your comment.

We recognized the deficiency in explaining various basic concepts in the original manuscript, prompting a systematic revision of the entire document. We are confident that in the current version, the interpretation of various concepts has been significantly clarified.

In the introduction part, we have incorporated a literature review of energy system security, placing emphasis on the innovative nature of the early warning system proposed in this paper.

So far, some research institutions, such as LANL, NREL and EMPA, have put forward theories related to energy security or resilience from different angles to assess or enhance the capacity of energy systems to sustain regular energy supply amidst disturbances. Grouped by disturbance sources, these theories can be classified as dealing with internal disruptions (e.g., gas leakage^{Error! Reference source not found.}) or external disruptions (e.g., extreme weather events^{Error! Reference source not found., Error! Reference source not found.}, cyber attacks^{Error! Reference source not found.}). Organized chronologically, they focus on the ability of energy systems to prepare for, absorb, adapt to, and recover from disruptive events before and after the disturbance^{Error! Reference source not found., Error! Reference source not found.}. In addition to traditional research on single energy resources, the exploration on the comprehensive resilience of coupled

energy systems has also made breakthroughs (e.g., hurricane analysis^{Error! Reference source not found.}, disaster avoidance^{Error! Reference source not found.}). However, most of the above studies play a limited role in the prevention of gas-electric cascading failures for the lack of real-time coordination between the two energy entities when the gas network malfunction occurs. This is related to the temporal misalignment of the failure concerning the two energy systems: the failure has already transpired in the gas system, while the power system is still on the verge of the failure. Current theories about energy resilience are less prone to take the advantage of this propagation delay in the medium, which has actually been demonstrated by the earthquake early warning system in the area of geology and disaster response^{Error! Reference source not found., Error! Reference source not found.}

In the 'Early Warning of the Gas System' section, we elucidate the physical basis of the gas-electric early warning system, drawing upon general principles of the early warning.

Early warning is defined as the perception of an impending danger, which is a form of situational awareness for the cyber-physical system. A comprehensive early warning system encompasses modules of sensing, analysis, decision-making, etc., intending to provide the response entity with more response time to minimize the losses^{Error! Reference source not found.}. In the area of disaster response, predicting the precise timing of disasters is challenging, but there exists correlation among them (e.g., an earthquake triggering a tsunami)^{Error! Reference source not found.}. This correlation involves a specific time interval surpassing the duration required by modules of the early warning system, rendering the early warning practicable in such instances. In coupled gas-electricity systems, there is such a correlation between the major failure of the gas network and the cascading failure of the power system caused by gas turbines. When the impacted gas turbines receive the control command from the EPCC, the cascading failure has not yet arrived at them.

In the 'Indicators of Gas System Failure' section, we have provided a more detailed explanation of the significance of early warning indicators, specifically in terms of sending complete fault information to the Electric Power Control Centre (EPCC) while safeguarding the privacy of the Gas Dispatch Centre (GDC). Additionally, we have revisited the definition of the two early warning indicators, AET and ALP.

In the gas-electric early warning system, certain indicators can be defined to help the power system escape from gas system failure, which are sent from the GDC to the EPCC. Here, we introduce two *early warning indicators*: the *available escape time* (AET) and the *available line pack* (ALP), which provide sufficient information of the gas network malfunction for power system control. Based on these two indicators, the EPCC can formulate a suitable control strategy for the power system without other gas network information. **Here, the GDC can indeed transmit comprehensive real-time information about the gas network to the EPCC, but in most countries, the management of coupled gas-electricity systems falls under two different entities, rendering this practice prone to privacy breaches.**

The AET is an indicator in the time dimension, which refers to the remaining time for the power system to escape from a gas system failure; in other words, this indicator denotes the time before the inlet pressure of the impacted gas turbine drops to the protection threshold according to *current* gas loads. Let the available escape time at time t be $AET(t)$. **If the generation of the gas turbine is decreasing while other gas loads remain constant, the inequation $AET(t) - AET(t + \Delta t) < \Delta t$ is satisfied, where Δt is the time interval.** This is due to the fact that the impacted gas turbine typically needs to decrease its power generation to escape from the failure, and the inlet pressure can be sustained for an extended period at a lower flow rate. If the AET is still positive when the power generation of the impacted gas turbine is controlled to zero, the gas system failure will not propagate to the power system through this turbine. To describe the emergency degree of the gas system failure itself, the initial AET (i.e. $AET(0)$) is defined as the *static available escape time* (SAET): the smaller the SAET, the more urgent the gas system failure is.

The location of the gas system failure relative to the impacted gas turbine is a key factor determining the SAET. In practice, if the impacted gas turbine is situated far from the failure location, it can continue operating for a long time before the inlet pressure drops to the threshold. To precisely measure this influence, we introduce the ALP as an indicator in the spatial dimension. The ALP refers to the gas line pack that remains available for the impacted gas turbine to operate without being forced to reduce the power generation after the gas system failure. As the impacted gas turbine is isolated from gas supply points, the line pack (denoted as LP) will decrease with the consumption of gas loads. When the current line pack fails to sustain the inlet pressure within the normal range, the gas turbine will be forced to reduce its generation output or even shut down under the action of low-pressure protection device. The line pack at this time is denoted as LP_{last} . As an early warning indicator, the ALP can be defined as the difference between LP and LP_{last} . Similarly, the initial ALP measures the urgency of the gas system failure when it just occurs.

We acknowledge that readers may perceive these concepts as highly abstract while navigating the 'Indicators of Gas System Failure' section. To enhance clarity, we have relocated the numerical example of the early warning indicators from the penultimate section to the end of the 'Indicators of Gas System Failure' section. This adjustment aims to foster a more cohesive understanding of the content.

In the 'Proactive Control of the Power System' section, we have also revised the definition of proactive control. (In the corresponding part in the Methods section, revisions have also been made.)

The formulation of proactive control strategies can be regarded as a special linear programming problem. The primary pursuit of proactive control is to minimize the total energy deficit during the control period. The constraints include power balance constraints, generator capacity constraints, ramp rate constraints, nonnegative ALP constraints, and maximum operating time constraints. Since the inlet pressure curve of gas turbines depends on the optimization results, the maximum operating time for each impacted gas turbine before the ALP declines to zero cannot be determined in advance. In practice, it can be set to the SAET in the first iteration — each impacted gas turbine is required to reduce its power generation to zero within its SAET. The proactive control method only containing the first iteration is called *static proactive control*. If the power deficit is fully eliminated under static proactive control, the iteration process is stopped; otherwise, the control time of the impacted gas turbines can be extended in subsequent iterations to further reduce the power deficit, which is called *dynamic proactive control*. The extension value can be determined according to the AET of each impacted gas turbine at the specified time in the last iteration. A detailed model analysis of this part is provided in the *Methods* section.

To delve deeper into the application scenarios of the gas-electric early warning system, we have expanded the case study section and introduced a numerical example for the Zhejiang Province system. This revision also makes up for the lack of numerical examples in the original manuscript and highlights the importance of the proposed method.

In order to verify the scalability of the gas-electric early warning system for large-scale systems, a case study of the provincial coupled gas-electricity system is also analyzed (shown in Figure 7). The provincial gas network adopts the structure of multiple gas sources, so it is difficult for a single major failure of gas sources or pipelines to have a substantial impact on gas supply. Therefore, we simulate simultaneous failures of multiple gas sources/pipelines during a winter storm. We present power generation curves for typical generators in the static proactive control strategy, including normal gas turbines, impacted gas turbines, coal-fired steam turbines, and nuclear power plants. In this strategy, the impacted gas turbines are required to control their power generation gradually to zero within their respective SAET, while other normal generators compensate for the power shortfall. Among these normal generators, nuclear power plants are set to bear the base load, so they are not involved in the power control process. Compared to swift adjustment of gas turbines, power generation curves of coal-fired steam

turbines exhibit greater stability due to the constraint of their complex physical structures. It is demonstrated that, in contrast to passive control, the energy deficit of the provincial system under this strategy is reduced from 512.0 MWh to 270.2 MWh within 2 hours after the failure.

In a word, we have meticulously organized and enhanced the theoretical framework encompassing the early warning system, early warning indicators, proactive control, and other related concepts. We believe that these improvements contribute to a clearer presentation of our work.

Main Comments (#3):

- 3. *The writing and presentation of the paper needs improvement. The figure captions are long and contain so much information. Fig. 1's data seems to date back to 2020. It would be better to have it updated with more recent information. Fig. 2 is confusing and the sequence does not quite show the cascade process. Also, some sentences are not adequately clear. For example, in page 5 in sentence "...suitable escape strategy of the power system without other gas system information, which effectively protects the privacy of the GDC.", what does the privacy of GDC refer to?*

Response:

Thank you for your valuable comments.

In this revised manuscript, we have reorganized the article's structure, enhanced explanations of key concepts, and refined the overall expression and grammar throughout the text.

In addressing the length of figure captions, we have consulted the journal's requirements and revised them accordingly to meet the stipulated criteria.

Concerning Figure 1, we have undergone a comprehensive revision to standardize the representation of the two subgraphs. Additionally, we have updated the gas pipeline length information to 2022, and the power generation information to 2020. These revisions incorporate the latest data available, obtained from relevant websites up to the present.

Fig. 1 | Development of natural gas utilities worldwide. **a**, Kilometres of gas pipelines by the end of 2022. The size of each sector on the left indicates the operating length of gas pipelines in a given country or region, while

the size of each sector on the right denotes the length of gas pipelines in development (proposed + construction), reflecting the growth momentum of gas infrastructure in this area. **b**, Global power generation and the proportion of natural gas. The size of each sector indicates the annual power generation in a given country or region, while the dashed area denotes the proportion generated by natural gas. **It can be seen that in the energy transition period, global natural gas utilities are basically in a growing momentum.** Data are retrieved from the literature Error! Reference source not found., Error! Reference source not found.

Concerning Figure 2, we have refined the presentation of the early warning system and extensively revised the figure captions. To accentuate the variation of key state variables when the early warning system is engaged, we have eliminated information unrelated to the principles of the gas-electric early warning system from the original figure.

Fig. 2 | Basic principle of the gas-electric early warning system. When a major failure occurs in the gas network, the adjacent valve chambers are closed to isolate the failure and the gas-electric early warning system starts to work. Three power generators (G0, G1, G2) are depicted to show the power generation adjustment in the control process, among which G1 and G2 are fired by the gas network. **a**, The major failure occurs in the gas network (I), and then failure signals are sent to the GDC (II). In this case, G2 is defined as the impacted gas turbine because its connection with the gas source is completely severed, and it can only consume the line pack in the terminal area. In the GDC, early warning indicators are calculated and sent to the EPCC (III). **b**, To reduce the impact of G2 on the power system, the EPCC formulates the proactive control strategy of the power system based on the early warning indicators (IV). The power generation of G2 is gradually reduced while that of other generators G0 and G1 is increased to maintain the balance of the power system (V). **c**, After a few minutes, if a complete substitution of

power generation by G2 is achieved before its inlet pressure drops to the protection threshold, the security of power supply will not be affected (VI).

We have also optimized expressions which were ambiguous. For instance, in addressing privacy concerns of the GDC, we have adopted a more concise and readily understandable statement and revised the statement as follows.

In the gas-electric early warning system, certain indicators can be defined to help the power system escape from gas system failure, which are sent from the GDC to the EPCC. Here, we introduce two *early warning indicators*: the *available escape time* (AET) and the *available line pack* (ALP), which provide sufficient information of the gas network malfunction for power system control. Based on these two indicators, the EPCC can formulate a suitable control strategy for the power system without other gas network information. Here, the GDC can indeed transmit comprehensive real-time information about the gas network to the EPCC, but in most countries, the management of coupled gas-electricity systems falls under two different entities, rendering this practice prone to privacy breaches.

At the end of the response, we thank you again for offering us so many valuable suggestions that help us improve our work. We sincerely hope that we have adequately addressed your concerns.

Response to the Reviewers: Reviewer 3

Comments to the Author:

- *The paper under review presents a method for provisioning a warning system that provides the coupled electricity grid with an early warning and malfunction data when an accident occurs in the natural gas network. The purpose of the warning system is to help the electricity grid avoid or reduce electric load shedding before the gas network line-pack is exhausted to the level where gas turbine power plants have to shut down immediately. This reviewer acknowledges the authors' efforts and conveys the following observations/comments.*

Response:

We sincerely thank you for your careful review and the valuable suggestions to improve the quality of the paper! According to your suggestions, we have made corresponding changes to the manuscript. The following are the answers to your questions and concerns. We hope that we have adequately addressed your concerns.

Main Comments (#1):

- *1. The first comment is: are the authors sure that there is no such a warning system in place now? It seemed highly improbable to the reviewer that gas operators do not provide related power plants with any warnings when the gas network is unable to supply the contracted gas. Maybe the authors are encouraged to review market models with respect to this kind of warnings and underlying proactive models.*

Response:

Thank you for bringing this matter to our attention, and we have taken it into careful consideration.

The 2021 winter storm in Texas highlighted instances where contracted gas supply faltered, and the gas-fired power plants involved were indeed notified in advance. However, a lack of coordination at the emergency control level resulted in unpreparedness at the EPCC when these plants shut down, leading to load shedding of the power system. In other words, the "warning system" for single gas-fired units does not significantly prevent the occurrence of major power outages. It is precisely such emergent scenarios that our method addresses.

In our approach, when the gas system fails, the global failure information of the gas system is sent by the GDC and then transmitted to the EPCC. Subsequently, the EPCC optimizes the output of each generator, reducing the likelihood of load shedding. The formulation of such a proactive control strategy needs global information of the system, which is hard to convey through single gas-fired units. As far as we know, the relevant research on the ancillary service or the reserve capacity market has not been able to solve the global control problem in such emergency scenarios (ref 1.1~1.3).

Moreover, even if the reserve capacity market continues to play the role in such emergency scenarios, our proposed approach can mitigate the need for reserve capacity, especially rapid-controlled reserve capacity, in the power system, thereby reducing the cost associated with reserve capacity. In passive control, the standby unit must ascend the slope at its maximum speed to minimize the impact of the power deficit on system frequency, demanding a high adjustment rate. Conversely, in proactive control, the power system calls these standby units for a longer time, so there is no need to prepare much rapid-controlled reserve capacity for the gas-electric cascading failure.

In order to better emphasize the theme of this article, we have reorganized the factual and rational arguments used in the introduction that power systems are more likely to be impacted by gas network failures, as follows:

However, compared to carbon emission reduction, the security of energy systems has not attracted enough attention from the public. With the expansion of gas networks, gas system malfunctions are now more likely to evolve into electric power outages. In February 2021, large-scale blackouts occurred in Texas and the south-central United States due to extremely cold weather^{Error! Reference source not found.}. Between February 8th and 20th, a gas generating capacity of 106,585 MW encountered unplanned outages or derates, primarily stemming from issues in the gas system^{Error! Reference source not found.}. **Despite there were advance notifications to each impacted gas turbine, the power system had to resort to load shedding in response to the crisis.** In the post-investigation report, the gas-electric interdependency was considered one of the most significant factors leading to the accident. **In fact, similar incidents took place in the United States in 1983, 1989, 2003, 2004, 2006, 2008, 2009, 2010, and 2011 due to adverse weather conditions**^{Error! Reference source not found.}. Additional factors such as pipeline leakage and illegal operations may also lead to gas-electric cascading failures, as exemplified in California (2015)^{Error! Reference source not found.} and Taiwan (2017)^{Error! Reference source not found.}. In short, power outages caused by gas network malfunctions are presently widespread and are anticipated to pose an increasing risk in the near future. Firstly, the transition from coal and oil to natural gas increases the dependence of electric power systems on gas networks, **and even the latter is gradually replaced by decarbonized gas fuels, e.g., blue or green hydrogen or ammonia, gas networks persist**^{Error! Reference source not found., Error! Reference source not found.}. Secondly, an increasing number of studies indicate that extreme weather events have become more frequent than before^{Error! Reference source not found., Error! Reference source not found.}, which may not only directly do harm to power systems but also threaten them by affecting the transportation of gas fuel^{Error! Reference source not found.}. Thirdly, global geopolitical tension amplifies the likelihood of human factors impacting the security of gas supplies, such as the Nord Stream gas pipeline explosion^{Error! Reference source not found.}.

Reference:

- 1.1 H. Farzin, R. Ghorani, M. Fotuhi-Firuzabad and M. Moeini-Aghtaie, "A Market Mechanism to Quantify Emergency Energy Transactions Value in a Multi-Microgrid System," in IEEE Transactions on Sustainable Energy, vol. 10, no. 1, pp. 426-437, Jan. 2019, doi: 10.1109/TSTE.2017.2741427.
- 1.2 A. Yousefi, E. Shayesteh, F. Daneshvar and M. P. Moghaddam, "A risk-based approach for provision of Spinning Reserve by means of Emergency Demand Response Program," 2008 IEEE 2nd International Power and Energy Conference, Johor Bahru, Malaysia, 2008, pp. 1011-1015, doi: 10.1109/PECON.2008.4762623.
- 1.3 A. Van Deventer, S. Chowdhury, S. P. Chowdhury and C. T. Gaunt, "Management of emergency reserves dispatching in electricity networks," 2010 International Conference on Power System Technology, Zhejiang, China, 2010, pp. 1-5, doi: 10.1109/POWERCON.2010.5666078.

Main Comments (#2):

- 2. *The reviewer recommends minor changes in the title of the article so that it can represent the purpose of the study in a clearer way. The article is NOT about an early warning, as the title suggests, but it explores a method of warning of failures in the natural gas network followed by a subsequent proactive control strategy.*

Response:

Thank you for your comment.

The technological advancements presented in this paper can be delineated into two key aspects. Firstly, we propose an early warning method for gas-electric cascading failures, centred around the identification

of the impacted gas turbines and the computation of early warning indicators by the GDC. Secondly, we propose the proactive control strategy based on these early warning indicators, with the primary focus being the regulation of each power generator by the EPCC.

Accordingly, we have incorporated your valuable suggestion and revised the title. Now the title of the paper is 'Early Warning and Proactive Control Strategies for Power Blackouts Caused by Gas Network Malfunctions'.

Main Comments (#3):

- 3. Line 15: "In a gas–electricity coupled system, the identification and transmission of gas network malfunction information using communication infrastructure occur significantly faster than power system dispatch". Why? What reference mentioned that? The sentence makes no sense.

Response:

Thank you for your valuable comment.

We have acknowledged the inaccuracy in our previous statement and clarified the applicable conditions for the early warning system. Presently, we posit that the early warning system is implemented under the condition that the time required for failure propagation from the gas system to the power system exceeds the sum of the time needed for identifying the gas failure, calculating the early warning indicators, and preparing the power system control. Given the inherently slow dynamics of gas systems, this condition is readily achievable.

The abstract is revised as follows.

There is growing consensus that gas-fired power generation will play a crucial role during the transition to net-zero energy systems, both as an alternative to coal- or petroleum-based power generation and as a flexibility service provider for power systems. However, malfunctions of gas networks have caused several large-scale power blackouts in the past. **The transition from coal and oil to gas fuels** significantly increases the interdependence between gas networks and electric power systems, thus raising the risks of more frequent and widespread power blackouts due to the malfunction of gas networks. In a coupled gas–electricity system, **the identification and transmission of gas network malfunction information, followed by the redispatch of electric power generation,** occur notably faster than the propagation and escalation of the malfunction itself, e.g., **significantly diminished pressure**. On this basis, we propose a gas-electric early warning system that can reduce the negative impacts of gas network malfunctions on the power system. An optimal proactive control strategy of the power system is also formulated based on the early warning indicators. The effectiveness of this method is demonstrated via case studies of a real coupled gas–electricity system in East China.

In our current revision, we have underscored the general principle of an 'early warning system.' Our core innovation lies in applying early warning principles, commonly employed in fields such as geology and disaster response, to multi-energy systems.

Main Comments (#4):

- 4. Line 13. "will significantly increase" is recommended instead of "significantly increases".

Response:

Thank you for your valuable comment.

Regarding the use of "increases" versus "will increase" in the abstract, we maintain our preference for the former. This choice is grounded in the acknowledgment that changes have been ongoing in the past decades and are expected to continue in the coming decades. Furthermore, it is precisely because of this

transition in the past that the sentence "malfunctions of gas networks have caused several large-scale power blackouts in the past" makes sense.

Main Comments (#5):

- 5. The reviewer believes that it would be helpful to explain in the introduction section that with penetration of more renewable generators, the natural gas network will have to counteract the intermittency of renewable generation, mainly through gas turbines. This duty for gas networks was not significant in the past. The new duty will make the gas networks more exposed to fluctuations, so a higher possibility of faults/failures can be expected. That is why gas networks will need much more accurate and sophisticated control programs.

Response:

Thank you for your valuable comment.

According to your suggestion, we have revised the first paragraph of the introduction as follows, in order to specifically highlight how gas turbines contribute to enhancing the penetration of renewable energy.

For electric power systems, it has become a common trend to gradually eliminate their dependence on fossil fuels and turn to low-carbon energy sources, in particular renewables such as wind and photovoltaics^{Error! Reference source not found., Error! Reference source not found.}. However, limited by the physical characteristic of power transmission and the intermittency of renewable generation, the development of conventional synchronous generators with sufficient capacity are still required that can be freely adjusted to maintain the balance between generation and the load of power systems^{Error! Reference source not found., Error! Reference source not found.}. Among all types of conventional generators, gas turbines have attracted public attention due to their superior performance and lower prices^{Error! Reference source not found.}. In contrast to coal- and oil-fired generators, gas turbines emit lower levels of carbon and pollutants and exhibit higher ramp rates which facilitates the integration of variable renewable resources^{Error! Reference source not found.,Error! Reference source not found.,Error! Reference source not found.}.

Main Comments (#6):

- 6. Lines 66 and 67. The reviewer expects a literature review on the security of coupled networks. However, only two references are mentioned insufficiently.

Response:

Thank you for your valuable comment.

Acknowledging your observation that the original text lacked sufficient presentation of existing work, we have now integrated the relevant content based on your suggestion. These studies combine studies on the resilience of single energy systems and integrated energy systems. The characteristics of the existing research are introduced to highlight the importance of our research.

So far, some research institutions, such as LANL, NREL and EMPA, have put forward theories related to energy security or resilience from different angles to assess or enhance the capacity of energy systems to sustain regular energy supply amidst disturbances. Grouped by disturbance sources, these theories can be classified as dealing with internal disruptions (e.g., gas leakage^{Error! Reference source not found.}) or external disruptions (e.g., extreme weather events^{Error! Reference source not found., Error! Reference source not found.}, cyber attacks^{Error! Reference source not found.}). Organized chronologically, they focus on the ability of energy systems to prepare for, absorb, adapt to, and recover from disruptive events before and after the disturbance^{Error! Reference source not found., Error! Reference source not found.}. In addition to traditional research on single energy resources, the exploration on the comprehensive resilience of coupled energy systems has also made breakthroughs (e.g., hurricane analysis^{Error! Reference source not found.}, disaster

avoidance^{Error! Reference source not found.}). However, most of the above studies play a limited role in the prevention of gas-electric cascading failures for the lack of real-time coordination between the two energy entities when the gas network malfunction occurs. This is related to the temporal misalignment of the failure concerning the two energy systems: the failure has already transpired in the gas system, while the power system is still on the verge of the failure. Current theories about energy resilience are less prone to take the advantage of this propagation delay in the medium, which has actually been demonstrated by the earthquake early warning system in the area of geology and disaster response^{Error! Reference source not found., Error! Reference source not found.}.

Main Comments (#7):

- 7. Line 52, Figure 1: “In this decade, although a general consensus has been reached on reducing global carbon emissions, ...”. Please note that in the reviewer’s opinion moving toward generating electricity from natural gas does not necessarily contradict the consensus on reducing greenhouse gases since natural gas-fired power plants produce less amount of greenhouse gases and other pollutants (with respect to other fossil-based power plants) in addition to paving the way for penetration of more renewable electricity, as it is also pointed out in the article. Maybe the authors want to correct the sentence by omitting the part that was mentioned in this comment.

Response:

Thank you for your valuable comment.

According to our research, the current development of natural gas is actually conducive to carbon emission reduction. The original statement did cause ambiguity, and we have removed the relevant statement in the current version.

Now, Figure 1 has been modified as follows.

Main Comments (#8):

- 8. Line 58. The authors are encouraged to provide a stronger reasoning to why power blackouts may become more probable in the future. They are recommended to mention that with the growth of hydrogen energy systems and the large-scale water electrolyzers coming into operation, the mutual energy exchange between gas networks and electricity grids will increase significantly, so a failure in one can have a severer impact on the other...

Response:

Thank you for your insightful guidance.

In response to your recommendations, we have incorporated examples of recent hydrogen storage projects in the UK, providing a more tangible and up-to-date context for our readers.

Many countries have built or are building large-scale gas networks and gas-fired electric power generation systems Error! Reference source not found., as shown in **Figure 1** Error! Reference source not found., Error! Reference source not found.. Moving towards carbon neutrality, although the use of natural gas as a fossil fuel will eventually be reduced, gas networks will be repurposed as important energy carriers of alternative sustainable fuels with energy storage and

transmission functionality^{Error! Reference source not found., Error! Reference source not found.}. For example, in early 2021, Shell, Mitsubishi Heavy Industries, Vattenwall and Wärme Hamburg signed a letter of intent to develop renewable hydrogen production in Hamburg, where the existing gas network will be expanded to accommodate the transport of hydrogen^{Error! Reference source not found.}. **In July 2023, UK Research and Innovation (UKRI) and Office of Gas and Electricity Markets declared a funding allocation of £95.3 million to push forward 10 large-scale demonstration projects, many of which aim to enhance the automation of gas systems and investigate the feasibility of integrating green hydrogen into the existing gas networks^{Error! Reference source not found.}.**

Regarding the discussion on the escalating frequency of gas-electric cascading failures, we have restructured the relevant paragraphs. We present factual arguments first, followed by rational arguments. Your point now serves as the first of our rational arguments.

However, compared to carbon emission reduction, the security of energy systems has not attracted enough attention from the public. With the expansion of gas networks, gas system malfunctions are now more likely to evolve into electric power outages. In February 2021, large-scale blackouts occurred in Texas and the south-central United States due to extremely cold weather^{Error! Reference source not found.}. Between February 8th and 20th, a gas generating capacity of 106,585 MW encountered unplanned outages or derates, primarily stemming from issues in the gas system^{Error! Reference source not found.}. **Despite there were advance notifications to each impacted gas turbine, the power system had to resort to load shedding in response to the crisis.** In the post-investigation report, the gas-electric interdependency was considered one of the most significant factors leading to the accident. **In fact, similar incidents took place in the United States in 1983, 1989, 2003, 2004, 2006, 2008, 2009, 2010, and 2011 due to adverse weather conditions^{Error! Reference source not found.}.** Additional factors such as pipeline leakage and illegal operations may also lead to gas-electric cascading failures, as exemplified in California (2015)^{Error! Reference source not found.} and Taiwan (2017)^{Error! Reference source not found.}. In short, power outages caused by gas network malfunctions are presently widespread and are anticipated to pose an increasing risk in the near future. Firstly, the transition from coal and oil to natural gas increases the dependence of electric power systems on gas networks, **and even the latter is gradually replaced by decarbonized gas fuels, e.g., blue or green hydrogen or ammonia, gas networks persist^{Error! Reference source not found., Error! Reference source not found.}.** Secondly, an increasing number of studies indicate that extreme weather events have become more frequent than before^{Error! Reference source not found., Error! Reference source not found.}, which may not only directly do harm to power systems but also threaten them by affecting the transportation of gas fuel^{Error! Reference source not found.}. Thirdly, global geopolitical tension amplifies the likelihood of human factors impacting the security of gas supplies, such as the Nord Stream gas pipeline explosion^{Error! Reference source not found.}.

Main Comments (#9):

- 9. Table one and also in the text: The reviewer recommends not assessing a power loss only based on the maximum megawatts. It would provide a much better understanding if the authors also provide the mean duration of power loss and the total electric energy loss (megawatt hours not supplied).

Response:

Thank you for your valuable suggestion.

Upon a comprehensive re-examination of accident reports, obtaining exact values for the two specified numbers has proven challenging. Firstly, not all reports explicitly provide information on total energy deficits, and secondly, variations in statistical methods further complicate obtaining uniform values.

Taking into account the feedback from reviewers, we recognize that the original table lacked sufficient cases to adequately illustrate the frequency of such incidents. As an alternative approach, we have decided to omit this table. Instead, we present a concise summary of events as a factual argument to underscore the

frequent occurrence of gas-electric cascading failures. We trust you understand our rationale for this decision.

Additionally, following your suggestion, we have supplemented the case studies with relevant information on the duration and total energy loss.

Main Comments (#10):

- 10. Line 127. Please explain why this inequality applies and what it means exactly.

Response:

This inequality underscores a fundamental characteristic of the AET, emphasizing its capability to sustain inlet pressure above the protection threshold for an extended duration when the output of the impacted gas turbine decreases. To illustrate, consider a scenario where the original endurance of a gas turbine at a certain output is 5 minutes. After 2 minutes, if the gas turbine's flow consumption has been diminishing during this period, the remaining endurance surpasses 3 minutes. This characteristic highlights the AET's feasibility in adapting to different gas turbine outputs over time.

The relevant statements are revised as follows.

The AET is an indicator in the time dimension, which refers to the remaining time for the power system to escape from a gas system failure; in other words, this indicator denotes the time before the inlet pressure of the impacted gas turbine drops to the protection threshold according to *current* gas loads. Let the available escape time at time t be $AET(t)$. If the generation of the gas turbine is decreasing while other gas loads remain constant, the inequation $AET(t) - AET(t + \Delta t) < \Delta t$ is satisfied, where Δt is the time interval. This is due to the fact that the impacted gas turbine typically needs to decrease its power generation to escape from the failure, and the inlet pressure can be sustained for an extended period at a lower flow rate. If the AET is still positive when the power generation of the impacted gas turbine is controlled to zero, the gas system failure will not propagate to the power system through this turbine. To describe the emergency degree of the gas system failure itself, the initial AET (i.e. $AET(0)$) is defined as the *static available escape time* (SAET); the smaller the SAET, the more urgent the gas system failure is.

Main Comments (#11):

- 11. Line 138. Maybe it is better to write “as the gas turbine is isolated from gas supply points, the line-pack will decrease....”.

Response:

We appreciate your guidance.

In response to your comment, we have revised the definition of the ALP.

The location of the gas system failure relative to the impacted gas turbine is a key factor determining the SAET. In practice, if the impacted gas turbine is situated far from the failure location, it can continue operating for a long time before the inlet pressure drops to the threshold. To precisely measure this influence, we introduce the ALP as an indicator in the spatial dimension. The ALP refers to the gas line pack that remains available for the impacted gas turbine to operate without being forced to reduce the power generation after the gas system failure. As the impacted gas turbine is isolated from gas supply points, the line pack (denoted as LP) will decrease with the consumption of gas loads. When the current line pack fails to sustain the inlet pressure within the normal range, the gas turbine will be forced to reduce its generation output or even shut down under the action of low-pressure protection device. The line pack at this time is denoted as LP_{last} . As an early warning indicator, the ALP can be

defined as the difference between LP and LP_{last} . Similarly, the initial ALP measures the urgency of the gas system failure when it just occurs.

Main Comments (#12):

- 12. The reviewer suggests adding “which can be regarded as the minimum escape time” after “in the first iteration”.

Response:

We appreciate your guidance.

Regarding this comment, it's crucial to clarify that SAET represents a minimum limit for the impacted gas turbine to escape, not the exact time required. Your suggestion prompted us to recognize the lack of clarity in our original expression. As a result, we have made systematic modifications to enhance clarity in the "Proactive Control of the Power System" section.

The formulation of proactive control strategies can be regarded as a special linear programming problem. The primary pursuit of proactive control is to minimize the total energy deficit during the control period. The constraints include power balance constraints, generator capacity constraints, ramp rate constraints, nonnegative ALP constraints, and maximum operating time constraints. Since the inlet pressure curve of gas turbines depends on the optimization results, the maximum operating time for each impacted gas turbine before the ALP declines to zero cannot be determined in advance. In practice, it can be set to the SAET in the first iteration — each impacted gas turbine is required to reduce its power generation to zero within its SAET. The proactive control method only containing the first iteration is called *static proactive control*. If the power deficit is fully eliminated under static proactive control, the iteration process is stopped; otherwise, the control time of the impacted gas turbines can be extended in subsequent iterations to further reduce the power deficit, which is called *dynamic proactive control*. The extension value can be determined according to the AET of each impacted gas turbine at the specified time in the last iteration. A detailed model analysis of this part is provided in the *Methods* section.

Main Comments (#13):

- 13. Line 183: The sentence has to be discussed in greater detail based on the Table 3 algorithm. The reviewer was unable to fully understand the algorithm because of the lack of explanation on parameters T_{set} and time-set ratio (α). Please clarify by adding new sentences.

Response:

We appreciate your guidance.

In the current 'Method' section, we have included a description of the two parameters. The parameter T_{set} is the time selected to calculate the new AET. The parameter time-set ratio α is an empirical parameter used to update the new control time, usually a number such as 0.1 or 0.2. The relevant paragraph has been revised as follows.

Superficially, the formulation of proactive control strategies is a linear programming problem, but the control time T_i is unknown in advance, making it difficult to solve accurately. Fortunately, the target of proactive control is to obtain a *good enough* control strategy within the shortest time and not pursue the *best* strategy. Therefore, the following iterative method can be used to obtain an adjustment strategy suitable for engineering practice. First, determine whether the control cost is acceptable in the *static proactive control strategy* where $T_i = SAET_i$ for each impacted gas turbine. If it is, this strategy can be used for power adjustment. Otherwise, the *dynamic proactive control strategy* can further be introduced to reduce the impact of the cascading failure, where the AET of each impacted gas turbine is recalculated according to the power generation and the ALP at $T_{set,i} = \alpha T_i$, where α is the

time-set ratio as an empirical parameter. The proactive control strategy can be reformulated by updating the proactive control time T_i with the new AET at $T_{set,i}$. This approach allows for the utilization of more reserve capacity in the power system and line pack in the terminal area, further reducing the adjustment cost until it reaches an acceptable level. The algorithm of dynamic proactive control is shown in Table 2.

Table 2 Algorithm of dynamic proactive control method.

Inputs	Early warning indicators of impacted gas turbines ($SAET_i$, $ALP_{0,i}$), parameters and variables of the power system
Parameters	Maximum iteration times ($iter_{max}$), control cost tolerance (tol), global control time (T), time-set ratio (α)
Step 1	Let the iteration number $k = 0$, and the proactive control time $T_i = SAET_i$ for each impacted gas turbine i .
Step 2	Solve the linear programming problem (9)-(17) and get the proactive control strategy ($P_{d,i,t}$, $P_{g,i,t}$) with the control cost $y_{c,k}$.
Step 3	Judge whether the control cost is acceptable or the remaining ALP for each impacted gas turbine is zero, that is, $y_{c,k} < tol$ OR $ALP_{i,T_i} = 0, \forall i \in S_{IG}$. If yes, go to the output step.
Step 4	Let $T_{set,i} = \alpha T_i$ and calculate $AET_{i,T_{set,i}}$ of each impacted gas turbine.
Step 5	Let $T_i = \max(AET_i(T_{set,i}), T_i)$.
Step 6	Let $k = k + 1$.
Step 7	Judge whether the iteration number $k < iter_{max}$. If yes, return to step 2.
Outputs	Proactive control strategy ($P_{d,i,t}$, $P_{g,i,t}$)

In fact, we have systematically reviewed the entire Method section to eliminate any potential ambiguity. We hope the current instructions are satisfactory.

Main Comments (#14):

- 14. Line 195: It is highly recommended to plot the diagram of the impacted power plant control time versus the ramp rate, showing the maximum operation time and the critical ramp rate. Furthermore, it is suggested to mention the capacity of the gas turbine under discussion and to omit the star sign from the labels in Figure 4 ($MW s^{-1}$ and $kg s^{-1}$ are enough).

Response:

We appreciate your guidance.

In fact, what you referred to as 'the impacted power plant control time versus the ramp rate' is already depicted in Figure b, as shown below. Control of the affected gas turbine concludes when the gas flow rate drops to zero. The bold red curve in the figure corresponds to the maximum control time. Additionally, we have removed the '*' symbol.

Figure 3 Change in key variables after gas system failure at $t = 0$.

Main Comments (#15):

- 15. Figure 4.d. The reviewer was unable to understand the descending behavior of the curves after the bold red curve (i.e. ramp rates above the critical ramp rate). It needs clarification.

Response:

We appreciate your comment.

In Figure 4, $AET(t)$ represents the time required for $ALP(t)$ to be depleted under the total gas mass flow at time t . In scenarios where the generation reduction ramp rate is greater than the critical point, the impacted gas turbine proactively shut down before the ALP is exhausted. Consequently, there is still a portion of ALP being consumed by the constant gas load, resulting in a declining curve.

To explain this, we have added the following sentence to the caption of Figure 3 (original Figure 4):

Fig. 3 | Change in key variables after gas system failure at $t = 0$. **a**, Schematic diagram of the terminal users of gas networks. When the major failure occurs, both the gas turbine and the ordinary gas load can only work with the gas stored in the blue-marked pipeline. The initial ALP for the impacted gas turbine is 5,000 kg. The initial mass flow rates of the gas turbine and the ordinary gas load are 5 kg/s and 3 kg/s, respectively. **b**, Mass flow curves of the gas turbine under different generation reduction ramp rates. The curve of the critical declining ramp rate is highlighted in bold red. When the ramp rate is lower, the gas turbine is faced with a sudden shutdown due to the insufficient inlet pressure. When the ramp rate is higher, the gas turbine is actively stopped before the inlet pressure drops to the threshold. **c**, ALP curves under different generation reduction ramp rates of the gas turbine. **d**, AET curves under different generation reduction ramp rates of the gas turbine. **If the ALP has not been used up when**

the gas turbine stops, the remaining portion will be consumed at a constant rate by the ordinary gas load.

Main Comments (#16):

- 16. Figure 4. It seems that “generation reduction ramp rate” is a more accurate label than “ramp rate”.

Response:

We appreciate your comment. Corrections have been completed.

Main Comments (#17):

- 17. Line 208 and similar cases. Please replace “gas-electricity coupling system” with “coupled gas-electricity system”.

Response:

We appreciate your comment. Corrections have been completed.

Main Comments (#18):

- 18. Figure 5. The label of pipelines is hard to read. Using alphabetic characters is suggested.

Response:

We appreciate your comment.

The size of the pipeline label has now been increased. We did not use alphabetic characters to avoid confusing the name of the pipeline with the English word when quoted in the article.

The revision of Figure 5 is as follows:

Main Comments (#19):

- 19. *Figure 6. Please mention the unit for unused ALP. In addition, it is highly recommended to plot power generation diagrams for the second case as well (failure of line 1). Also, the authors may want to report the duration of power deficit and the total load shed as comparison parameters in a table for all of the 5 cases.*

Response:

We appreciate your comment.

The unit of unused ALP (kg) is already indicated on our axes. Regarding the typical power control curves, we have adjusted the corresponding scenario to the second case for a more effective comparison. Prioritizing visual aesthetics, we have chosen the maximum power deficiency and unused ALP as key outcomes in power system control. Our objective is to illustrate the gradual elimination of power deficits with different utilization efficiency of the ALP. While incorporating duration and energy loss for each of the five cases might lead to visual clutter due to overlapping axes, your suggestion has inspired us to annotate the duration and energy loss in case 2, which is shown as follows.

Main Comments (#20):

- 20. Line 222. Please add the word “after” after the word “shut down”.

Response:

We appreciate your comment. Corrections have been completed.

Taking the failure of the gas source (GS) as an example, if the EPCC does not adjust the generation of any power plant, gas turbines XS and HZ will be forced to shut down 30 min 25 s and 13 min 21 s **after** the failure occurs in the gas system, resulting in electric power deficits of 1,047.2 and 675.4 MW, respectively.

Main Comments (#21):

- 21. Last case in Figure 6. Please explain why such a short control time is adopted for the power plant

HZ (00:01:38 and 00:01:07) while in Line 223 it was mentioned that HZ has a SAET of 00:13:21 (13 minutes and 21 seconds)

Response:

We appreciate your comment.

In our analysis of the seven-pipeline case study, we systematically examine the implications of gas source failure and the failure of each pipeline, calculating the early warning indicators for these N-1 failures. On original line 223, we specifically address the gas source (GS) failure. In this scenario, with the abundant ALP in the terminal area, each gas turbine can persist for a longer period. Contrasting this, the final case illustrated in Figure 6 pertains to a failure in pipeline ⑤. In this instance, gas turbine HZ is constrained to utilize the gas stored in pipeline ⑥, resulting in a shorter endurance period.

To make the explanation clearer, we have rearranged the paragraphs on the early warning and the proactive control in the numerical example:

The application of the proposed proactive control method is demonstrated with the coupled gas–electricity system depicted in Figure 5. Both the gas system and the electric power system are part of the actual system in Zhejiang Province. For the sake of simplicity, the coupled gas–electricity system in one major city is selected for main verification, and it is assumed that the system operates in an isolated mode (i.e., the inter-city support is not considered). According to the topology of the gas network, certain major failures (such as failures of the gas source and pipelines ①, ③, ④, and ⑤) will impact the electric power system ($SAET < \infty$), while other failures (such as pipelines ②, ⑥, and ⑦) will not ($SAET \rightarrow \infty$), depending on whether the fuel supply of gas turbines will be severed. Given each major failure in the gas system, the SAET of impacted gas turbines is calculated by the GDC (shown in Figure 5). Taking the failure of the gas source (GS) as an example, if the EPCC does not adjust the generation of any power plant, gas turbines XS and HZ will be forced to shut down 30 min 25 s and 13 min 21 s after the failure occurs in the gas system, resulting in electric power deficits of 1,047.2 and 675.4 MW, respectively.

Under the failure of GS, the power deficit can be fully eliminated once the proactive control method is adopted by the EPCC (shown in Figure 6). As the SAET of each impacted gas turbine is long enough for the EPCC to substitute its power generation by other normal generators, the proactive control strategy is not iterated. In other words, only the static proactive control strategy is required to reduce the power deficit to zero. Under the failure of each pipeline, the proactive control strategy can also be formulated by the EPCC based on the AET and the ALP to reduce the cascading impacts on the power system. With the help of the early warning system, the power deficit under each pipeline failure can be significantly reduced. In particular, not all contingencies in the gas system provide a sufficient SAET for the EPCC to escape from the cascading failure. For instance, as the distance from HZ to the nearer isolation valve decreases, the SAET of HZ decreases monotonically under the failures of pipes ③, ④ and ⑤ in sequence. The calculation demonstrates that under the failure of pipes ③ and ④, the EPCC can maintain the power balance using the spare capacity within the SAET, while under the failure of pipe ⑤ it cannot—if the power generation of HZ must be reduced to zero within the SAET, the power system will lose 507.7 MW of power generation. However, since the SAET suggests a constant power generation of the gas turbine, when this power generation declines, the maximum operating time of the gas turbine will also increase. In the dynamic proactive control scenario, if the control time of HZ is extended from 1 min 7 s to 1 min 38 s, the power deficit will be reduced maximally from 507.7 to 438.4 MW with the ALP exhausted. This reduction could be more substantial if the HZ was equipped with gas storage facilities to provide more line pack.

Main Comments (#22):

- 22. Line 226. How is it proved that EPCC had enough time and available reserve capacity to substitute

the lost generations?

Response:

We appreciate your comment.

This sentence still refers to the case of gas source failure. As explained above, the available gas line pack and endurance time of each gas turbine are very long when the gas source fails. In our calculation, the power deficit in this case can be completely eliminated by adopting static proactive control, which shows that the control time and reserve capacity are sufficient.

Main Comments (#23):

- 23. Line 227. *According to the provided algorithm in the Methods section, the word necessarily is not required here.*

Response:

We appreciate your comment. Now we have corrected this sentence.

Main Comments (#24):

- 24. Line 292. *The reviewer believes “unknown” is more accurate than “uncertain”.*

Response:

We appreciate your comment. Corrections have been made.

Main Comments (#25):

- 25. Line 310. *References are required.*

Response:

We appreciate your comment. Corrections have been made.

Main Comments (#26):

- 26. Line 314. *Please provide references for equations (1).*

Response:

We appreciate your comment.

Telegraph equations can be found in many textbooks on circuit principles, and for our reference, we have chosen a classic paper in which multi-energy systems are simulated based on the energy-circuit theory.

Main Comments (#27):

- 27. Line 331. *Please support this hypothesis with underlying gas dynamic equations.*

Response:

Thank you very much for your comment. We have noticed some inaccuracies in the original statement and have made some modifications. The basis for this hypothesis is the mass conservation, the momentum conservation and the gas state equations, which together constitute the isothermal model of the gas pipeline commonly used in the industrial field (ref 1):

$$\frac{\partial \rho}{\partial t} + \frac{\partial(\rho v)}{\partial x} = 0$$

$$\frac{\partial(\rho v)}{\partial t} + \frac{\partial(\rho v^2)}{\partial x} + \frac{\partial p}{\partial x} + \frac{\lambda \rho v |v|}{2D} + \rho g \sin \theta = 0$$

$$p = ZRT\rho$$

The independent variable includes time coordinate t and space coordinate x , the dependent variable includes density ρ , flow speed v and pressure p , and the parameters include pipeline friction coefficient λ , inner diameter D , gravity acceleration g , slope angle θ , compression factor Z , gas constant R and temperature T . Long-distance transmission pipeline usually works in the resistance square area, when the friction coefficient λ is only related to the pipeline parameters, not influenced by the Reynolds number. By substituting variables, we can get the form of Eq. (1) in the manuscript (telegraph equations), which is essentially a wave equation with the propagation speed $c = \sqrt{ZRT}$.

The exact expression of our assertion is that in the absence of gas flow within a pipe, encompassing both the gas source and the gas load, the pressure across various points will tend to equalize. This phenomenon is better understood considering that the speed of gas pressure wave is approximately the speed of sound (about 300 m/s), which can be proved by expressing the dynamic equations of gas pipelines in the form of a wave equation (ref 2). Gas pressure waves undergo multiple reflections in the gas pipeline over a short period, leading to a tendency for uniform pressure at each point along the pipeline.

In our examination of the terminal area linked to the gas turbine, the AET, denoting the time for the gas turbine to cease operation due to insufficient pressure, is a key factor in our optimization model. The model reveals that the mass flow of gas turbines is usually reduced steadily in the proactive control strategy. (In real passive control of impacted gas turbines, there is still a gradual decline in the mass flow, which is omitted in our Figure 4 for the sake of simplicity.) When the gas turbine undergoes shutdown, its impact on the gas flow within the pipeline is typically minimal, validating the plausibility of this hypothesis. As an illustration, consider the gas pipeline depicted in the figure as follows. Let's assume that the gas supply is lost at the head end, transforming it into a point where the flow boundary condition is zero. Subsequently, the flow rate of the terminal gas turbine decreases at a rate of 5% capacity per minute. The accompanying figure below depicts the pressure curve for each point along the pipeline at different times, and the table shows the standard deviation of pressure per kilometre. That is to say, after the failure of the gas source, the pressure in the pipeline tends to be uniform.

Fig. Parameters of the verification case

Fig. Pressure curves at different times

Table Standard deviation of pressure per kilometre at different times

Time / min	0	5	10	15	20
Standard deviation / bar	0.1332	0.03207	0.01212	0.0012	0.0045

The situation becomes more intricate when there is a constant gas load in the terminal area. If the shutdown pressure of the constant gas load is lower than that of the gas turbine, gas flow inside the pipeline persists when the gas turbine is shut down. However, there is typically a certain distance between the gas turbine and other gas loads. Consequently, the pressure valley in the terminal area will also be situated at a distance from the gas turbine, generally preventing the pressure at the gas turbine location from dropping below the average pressure across the entire area. This usually results in a larger estimate of the final line pack and a smaller estimate of the available line pack. Despite leading to a smaller estimate of the AET, indicating a more 'urgent' failure determination, it still aligns with the requirements of proactive control. As an illustration, consider the gas network depicted in the figure as follows. Let's assume that the gas supply is lost at the head end. Subsequently, the flow rate of the terminal gas turbine decreases at a rate of 5% capacity per minute. The accompanying figure below depicts the pressure curve for each point in the network at different times, and the table shows the ratio between the inlet pressure of the gas turbine and the average pressure of the whole network. It can be seen that after the gas source failure occurs, the error between the gas turbine inlet pressure and the mean pressure of the network is very small and decay rapidly, so it is feasible to estimate the final line pack with the minimum inlet pressure of the gas turbine.

Fig. Parameters of the verification case

Fig. Pressure curves at different times

Table Pressure ratio between Node 2 and the average of the whole network

Time / min	0	5	10	15	20
Pressure ratio / bar	0.9691	0.9805	0.9918	0.9982	1.0006

We also compare the SAET calculated under this hypothesis with the time needed for gas turbine inlet pressure to drop to the threshold through the dynamic simulation method. The figure below illustrates the topology of the gas network, while the table provides the computation outcomes of the early warning time for various branch failures. The control pressure of the gas source is set at 2 MPa. The constant gas loads are 5.98, 4.98, and 2.99 kg/s in turn. The output of each gas turbine is 80 MW with a thermal efficiency of 0.55, and both of them have a minimum inlet pressure limit of 1.5 MPa. Branch ⑧ is a compressor with an output pressure of 2 MPa and a compression ratio ranging from 1.1 to 2.0.

Fig. Topology and parameters of the verification case

Table Early warning time under the failures of pipeline ①⑤⑥

Pipeline serial number		①	⑤	⑥
SAET	G1	Calculated under the hypothesis	15.68	/

(min)		Dynamic hydraulic simulation	15.10	/	/
	G2	Calculated under the hypothesis	57.63	30.48	11.14
		Dynamic hydraulic simulation	58.57	30.50	11.17

In essence, our introduction of early warning indicators aims to 'condense' the information of the gas network malfunction within an acceptable margin of error. This method provides the proactive control parameters for the power system in a comprehensible manner without significantly compromising the privacy of the gas system. If we want to make the early warning more accurate, it is necessary to introduce a more complete dynamic hydraulic analysis of the gas network, and update the boundary conditions of coupling points iteratively. For proactive control after the failure, this method is difficult to meet the time requirement.

Reference:

1. Zhu, G.Y., Henson, M.A., Megan, L., 2001. Dynamic modeling and linear model predictive control of gas pipeline networks. *J. Process Control* 11, 129-148.
2. B. Chen et al. Energy-Circuit-Based Integrated Energy Management System: Theory, Implementation, and Application. *Proceedings of the IEEE* 110, 1897-1926 (2022). <https://doi.org/10.1109/JPROC.2022.3216567>.

Main Comments (#28):

- 28. Line 406, The sentence beginning with “otherwise” needs more explanations and discussion.

Response:

We appreciate your comment. Corrections have been made, which have been shown in previous responses #13.

Main Comments (#29):

- 29. Table 3. Please discuss the time-set ratio.

Response:

We appreciate your comment. Corrections have been made, which have been shown in previous responses #13. In fact, the time-set ratio is an empirical parameter used to calculate the new control time, usually a number such as 1.05 or 1.1.

Main Comments (#30):

- 30. A medium size case study has the ability to better show the effectiveness of the method discussed in this paper.

Response:

We appreciate your comment.

We have included a large-scale numerical example featuring a province-wide network, focusing on the study of typical generation control curves of different types of generators such as normal gas turbines, impacted gas turbines, coal-fired steam turbines, and nuclear power plants, which is shown in Figure 7.

In order to verify the scalability of the gas-electric early warning system for large-scale systems, a case study of the provincial coupled gas-electricity system is also analyzed (shown in Figure 7). The provincial gas network adopts the structure of multiple gas sources, so it is difficult for a single major failure of gas sources or pipelines to have a substantial impact on gas supply. Therefore, we simulate simultaneous failures of multiple gas sources/pipelines during a winter storm. We present power generation curves for typical generators in the static

proactive control strategy, including normal gas turbines, impacted gas turbines, coal-fired steam turbines, and nuclear power plants. In this strategy, the impacted gas turbines are required to control their power generation gradually to zero within their respective SAET, while other normal generators compensate for the power shortfall. Among these normal generators, nuclear power plants are set to bear the base load, so they are not involved in the power control process. Compared to swift adjustment of gas turbines, power generation curves of coal-fired steam turbines exhibit greater stability due to the constraint of their complex physical structures. It is demonstrated that, in contrast to passive control, the energy deficit of the provincial system under this strategy is reduced from 512.0 MWh to 270.2 MWh within 2 hours after the failure.

Fig. 7 | Provincial demonstration of the gas-electric early warning system. This figure shows power generation curves of different types of generators given multiple gas major failures in an external extreme event, including normal gas turbines (GT1, GT2), impacted gas turbines (GT3, GT4), coal-fired steam turbines (ST1, ST2) and nuclear power plants (NP1, NP2). Generally, gas turbines are more capable of regulating than coal-fired steam turbines, and nuclear power plants are usually not involved in regulating. Typically, gas turbines exhibit greater regulatory capabilities compared to coal-fired steam turbines, while nuclear power plants are rarely involved in the generation adjustment process.

Main Comments (#31):

- 31. For the fastest expansion of knowledge, the reviewer suggests making the data related to case studies available to other researchers.

Response:

We appreciate your comment. Corrections have been made, which is shown in Data Availability.

Main Comments (#32):

- 32. All the parameters and variables that are used in equations must be introduced.

Response:

We appreciate your comment.

Corrections have been made. We combed through the text in the "Method" section to make sure that each variable was defined before it was used.

Main Comments (#33):

- 33. *The equations related to gas pressure are not valid in general. So for all equations, authors must discuss governing assumptions.*

Response:

We appreciate your comment. In the current "Methods" section, we have made more efforts to point out the governing assumptions in pressure calculation. We have also included more references to enhance the transparency and credibility of our methodology.

The first assumption is the isothermal model of gas pipelines, which is a widely used mathematical framework in numerous studies investigating the dynamics of gas systems (ref 1~3). The more comprehensive dynamic model of gas pipelines typically comprises three partial differential equations governing the conservation of mass, momentum, and energy, along with an algebraic equation characterizing the gas state. This broader model explores the spatiotemporal variations of four physical quantities: pressure, density, velocity, and temperature. However, the isothermal model simplifies this complexity by neglecting the temperature change resulting from gas transmission. Additionally, through the linear pressure-density relationship provided by the gas state equation, we derive the expression represented by Eq. (1) in our manuscript. This derivation involves the selection of pressure and mass flow as the chosen state variables.

The second assumption is to consider only the dynamic process of the natural gas pipeline and ignore the dynamic process of other components in the system. This modeling approach, commonly referred to as the quasi-dynamic model, also finds application in many studies (ref 4~5). Given that the dynamic adjustments of components such as valves and compressors typically require only a few minutes to achieve a steady state, in contrast to the tens of minutes or hours required for gas transmission in the pipeline, which means that this simplification does not significantly impact the accuracy of calculations.

The third assumption is the convergence of the iterative difference scheme Eq. (4). The convergence is assured by the Lax equivalence theorem, stating that if the corresponding difference equation adheres to the consistency condition for the definite solution of a well-posed linear system of partial differential equations, then the sufficient and necessary condition for the convergence of the difference equation is the stability of the scheme (ref 6). As Δx and Δt approach zero, the truncation error of linear approximations (2) and (3) in the manuscript tends to zero, ensuring the consistency condition. Additionally, it can be demonstrated that the difference scheme remains stable when the following two equations are satisfied, thereby confirming the convergence of the iterative difference scheme:

$$1 + 4 \frac{\Delta t}{L_0 \Delta x} \leq C$$
$$1 - \frac{R_0 \Delta t}{L_0} + \frac{2 \Delta t}{C_0 \Delta x} + \frac{4(\Delta t)^2}{L_0 C_0 (\Delta x)^2} \leq C$$

The fourth assumption is that each point in the gas network where there is no gas flow at the boundary will soon reach approximately the same pressure. This assumption has been discussed in reply #27. The main purpose of this assumption is to quickly calculate the early warning indicators which 'condense' the information of the gas network malfunction within an acceptable margin of error.

Reference:

1. Zhu, G.Y., Henson, M.A., Megan, L., 2001. Dynamic modeling and linear model predictive control of gas pipeline networks. *J. Process Control* 11, 129-148.
 2. B. Chen et al. Energy-Circuit-Based Integrated Energy Management System: Theory, Implementation, and Application. *Proceedings of the IEEE* 110, 1897-1926 (2022).
 3. Osiadacz, A. J., & Chaczykowski, M. (2001). Comparison of isothermal and non-isothermal pipeline gas flow models. *Chemical Engineering Journal*, 81(1-3), 41-51.
 4. Behrooz, H. A., & Boozarjomehry, R. B. Modeling and state estimation for gas transmission networks. *Journal of Natural Gas Science and Engineering* 22, 551-570 (2015).
 5. Jenicek, T., & Kralik, J. (1995, October). Optimized control of generalized compressor station. In *PSIG annual meeting* (pp. PSIG-9502). PSIG.
 6. Tekriwal, M., Duraisamy, K., & Jeannin, J. B. (2021, May). A formal proof of the Lax equivalence theorem for finite difference schemes. In *NASA Formal Methods Symposium*(pp. 322-339). Cham: Springer International Publishing.
-

Main Comments (#34):

- 34. *Fig 3: the passive control: is not like illustrated in the figure, because the gas pressure sensors at the power plants can capture the pressure drop immediately at the time of gas pipe failure. You may be interested to know that they can even estimate the location of pipeline failure with an acceptable precision.*

Response:

Thank you for your insightful question. We have realized the limitations of original Figure 3.

In the event of a large-scale accident, such as the one in Texas in 2021, individual gas plants typically receive advance information about the impending gas supply shortage. However, this information exchange occurs between the Gas Dispatch Centre (GDC) and individual gas plants, rather than between the GDC and the Electric Power Control Centre (EPCC). As a consequence, different power plants may exhibit distinct behavioural patterns—some may opt to gradually reduce their output before shutting down, while others may choose to shut down directly. This also leads to a number of situations to draw a passive control curve rigorously.

The passive control depicted in Figure 4 (original Figure 3) corresponds to the system illustrated in Figure 2. Essentially, it showcases the scenario where G2 shuts down directly after maintaining the original power for the longest time. This scenario represents the simplest case aligning with real-world dynamics, which provides a benchmark for the comparison between passive control and proactive control.

To express this point more clearly, we have revised the caption of Figure 4 (original Figure 3) as follows:

Fig. 4 | Comparison of power control processes between passive control and proactive control. To illustrate the control process, the initial power generation levels of G0, G1 and G2 are selected as typical engineering values. **a**, Scenario of the passive control strategy. When the gas network failure occurs, G2 first tries keeping its original power generation for as long as possible. After the SAET, it will shut down due to insufficient inlet pressure, resulting in a power deficit. In engineering practice, if such a power deficit is large enough, it may cause load shedding or even collapse of the power system. **b**, Scenario of the proactive control strategy. G2 is required to reduce its power generation to zero before the ALP runs out, while G0 and G1 can fully offset the power deficit in real time, so there is no fluctuation in the power system. As the power generation of G2 is reduced, its maximum operation time exceeds the SAET.

In fact, your suggestion actually provides us with a good research prospect, that is, the power system can realize the localized early warning of gas-electric cascading failure through a small number of measurements of the gas system. This could potentially involve coordinating the pressure and flow measurements from various gas-fired units.

Main Comments (#35):

- 35. Line 212: as the gas runs through a network of pipes, not a single pipe, it is possible that the pressure drop due to pipeline failure might be compensated by other lines connected to each other. Therefore the one line assumption is too simplistic in a way that change the nature of the event.

Response:

Corrections have been made.

In our new medium-scale case study shown in Figure 7, we have incorporated validation of provincial gas networks with N-m failures. The outcomes affirm the continued applicability of the early warning system and proactive control method even when addressing N-m failures within large-scale systems. Consequently, our example now encompasses both a small isolated network under N-1 failure and a large interconnected network under N-m failure, contributing to the enhanced credibility of the numerical examples.

Main Comments (#36):

- 36. Line 81: the frequency of events (gas network failure) is 4 times in 9 years, all over the world and it has not been repeated for any of locations. It shows that local EPCCs might figure out a way to solve the problem or basically it is not an important issue to address.

Response:

Thank you for your valuable advice.

Acknowledging the limitations in the types of failures initially listed in original Table 1, we have rephrased and presented them in a different manner. The intent of original Table 1 was to convey the increasing prevalence of gas-electric coupling accidents. However, the original list of four failures proved insufficient in illustrating this point effectively. In the revised version, we opted for a textual expression, enumerating 11 failures. This approach allows us to emphasize the distinctive characteristics of these failures, showcasing their wide occurrence and high repeatability. This modification underscores the significance of the work presented in this paper.

However, compared to carbon emission reduction, the security of energy systems has not attracted enough attention from the public. With the expansion of gas networks, gas system malfunctions are now more likely to evolve into electric power outages. In February 2021, large-scale blackouts occurred in Texas and the south-central United States due to extremely cold weather^{Error! Reference source not found.}. Between February 8th and 20th, a gas generating capacity of 106,585 MW encountered unplanned outages or derates, primarily stemming from issues in the gas system^{Error! Reference source not found.}. Despite there were advance notifications to each impacted gas turbine, the power system had to resort to load shedding in response to the crisis. In the post-investigation report, the gas-electric interdependency was considered one of the most significant factors leading to the accident. In fact, similar incidents took place in the United States in 1983, 1989, 2003, 2004, 2006, 2008, 2009, 2010, and 2011 due to adverse weather conditions^{Error! Reference source not found.}. Additional factors such as pipeline leakage and illegal operations may also lead to gas-electric cascading failures, as exemplified in California (2015)^{Error! Reference source not found.} and Taiwan (2017)^{Error! Reference source not found.}. In short, power outages caused by gas network malfunctions are presently widespread and are anticipated to pose an increasing risk in the near future.

Main Comments (#37):

- 37. Figure1-a,b: bad illustration. Hard to understand at the first sight and it is also hard to interpret after even reading the caption. In fact, it is not following the best practice of data visualization.

Response:

Thank you for your valuable advice.

It has been corrected. We hope you enjoy our current figure, which has also been updated with the latest data.

Response to the Reviewers: Reviewer 4

Comments to the Author:

- *This paper develops a method to better deal with the failures of the gas network that can propagate to the power system and cause outages and blackouts. The paper computes two metrics: available escape time (AET) and available line pack (ALP) that are communicated to the power system control center, as soon as a failure in the gas system occurs. These metrics are integrated within an optimization model that produces proactive control strategies, to reduce the negative impacts on the power system.*
- *The increasing interdependence between the gas and power systems is an important problem, adding to the complexity of reliable operation of the two systems. However, it is not clear if the method proposed in the paper is sufficient for addressing this challenging problem. I suggest that authors add a subsection to clearly describe and discuss the particular applications of their proposed method. For instance, clearly state the type of events, where the method is useful and the events where the method would not be so useful. Additionally, a discussion of the strengths and weaknesses/shortcomings seems to help the readers better understand the scope of the work and its applications.*

Response:

We sincerely thank you for your careful review and the valuable suggestions to improve the quality of the paper! Following your suggestion, we have incorporated a Discussion section to thoroughly explore the applicability, advantages, and disadvantages of the method proposed in this paper. The following are the answers to your questions and concerns. We hope that we have adequately addressed your concerns.

Main Comments (#1):

- *1. Consider adding a discussion about N-1 reliability standards and their shortcomings relative to the work presented in this paper.*

Response:

Thank you for your valuable advice.

In our original examples, we actually utilized the N-1 principle to define major failures. For now, however, we have expanded our examples to include N-m scenarios of large-scale systems, demonstrating the scalability of the proposed method.

In order to verify the scalability of the gas-electric early warning system for large-scale systems, a case study of the provincial coupled gas-electricity system is also analyzed (shown in Figure 7). The provincial gas network adopts the structure of multiple gas sources, so it is difficult for a single major failure of gas sources or pipelines to have a substantial impact on gas supply. Therefore, we simulate simultaneous failures of multiple gas sources/pipelines during a winter storm. We present power generation curves for typical generators in the static proactive control strategy, including normal gas turbines, impacted gas turbines, coal-fired steam turbines, and nuclear power plants. In this strategy, the impacted gas turbines are required to control their power generation gradually to zero within their respective SAET, while other normal generators compensate for the power shortfall. Among these normal generators, nuclear power plants are set to bear the base load, so they are not involved in the power control process. Compared to swift adjustment of gas turbines, power generation curves of coal-fired steam turbines exhibit greater stability due to the constraint of their complex physical structures. It is demonstrated that, in contrast to passive control, the energy deficit of the provincial system under this strategy is reduced from 512.0 MWh to 270.2 MWh within 2 hours after the failure.

Fig. 7 | Provincial demonstration of the gas-electric early warning system. This figure shows power generation curves of different types of generators given multiple gas major failures in an external extreme event, including normal gas turbines (GT1, GT2), impacted gas turbines (GT3, GT4), coal-fired steam turbines (ST1, ST2) and nuclear power plants (NP1, NP2). Generally, gas turbines are more capable of regulating than coal-fired steam turbines, and nuclear power plants are usually not involved in regulating. Typically, gas turbines exhibit greater regulatory capabilities compared to coal-fired steam turbines, while nuclear power plants are rarely involved in the generation adjustment process.

Besides, in the Discussion section, we pinpoint the limitations of existing research on power system security based on the N-1 principle. We also emphasize the advantages inherent in the method presented in this paper.

With the development of natural gas and decarbonized gas fuels, the power system's dependence on the gas transmission network is steadily increasing. In such a scenario, traditional theories of power system security, especially those based on the N-1 principle, face limitations. Real-world experiences of power outages highlight that extreme external events can disrupt fuel supply to multiple gas turbines, rendering security protection measures based on the N-1 principle ineffective. In response, we draw inspiration from the early warning principles successfully applied in geology and disaster response, and propose the gas-electric early warning system. This innovative system offers extended time for adjusting generator outputs, aiming to minimize the emergency control cost and reduce the likelihood of power outages.

Main Comments (#2):

- 2. Elaborate why and if AET and ALP provide sufficient information for power system operation, regarding the failures in the gas network. If not, what sort of additional information may be needed?

Response:

Thank you very much for your advice.

The AET and the ALP serve as an equivalent representation of the terminal pipeline segment, offering

a comprehensive description of the gas system failure for the EPCC. These two indices, in conjunction, fulfil the requirements for proactive control of the power system.

Taking SAET and ALP(0) as exemplars, if the EPCC possesses these two indicators, and given its mastery of the output and flow consumption of the impacted gas turbines, the EPCC can calculate the variation of ALP(t) using the generation curve of the gas turbine. Moreover, as evident from the calculation process of the early warning indicators outlined in the Methods section, the ordinary gas load component within the terminal area can be effectively calculated the using two early warning indicators. Therefore, this approach guarantees that the optimization problem does not necessitate additional input parameters.

In order to better show the advantages and limitations of the early warning system, we have also incorporated additional content into the Discussion section.

In the gas-electric early warning system, the GDC analyses global information regarding gas major failures and notifies the EPCC in advance, so that the latter can systematically optimize the operation of the power system. The quantification of this global information is achieved through early warning indicators such as the AET and the ALP. In our study, these two indicators can provide sufficient information for the proactive control of the power system, eliminating the need for the EPCC to access more detailed information about the gas system. This underscores that the gas-electric early warning system does not simply combines the information of the two systems; it meets the demand of privacy protection across multiple entities.

The basic premise of the gas-electric early warning system is that when a major failure occurs in the gas system, the storage capacity of gas pipelines serves as a buffer for failure propagation. If this buffer is not enough to maintain the inlet pressure, the effectiveness of the early warning system will be limited. In practical engineering applications, it is possible to configure typical failure scenarios within the gas system by considering the failure probabilities of various components. This assessment serves to evaluate whether the storage capacity aligns with the requirements of the gas-electric early warning system. If inadequacies are identified, additional gas storage facilities can be installed to further ensure the feasibility of the gas-electric early warning system, thereby enhancing the resilience of coupled gas-electric system to external extreme events.

Main Comments (#3):

- 3. *The authors seem to have ignored a considerable body of literature that aims to solve a similar problem. At least a brief literature review, comparing this paper to the existing literature is required to clarify the contribution of the work. Look, for instance, at Los Alamos National Laboratory's work on power and gas networks.*

Response:

Thank you for your valuable advice.

We have incorporated relevant literature reviews in the Introduction section, including the work in various laboratories. Existing studies have concentrated on the periods before, during, and after failures, yet overlooked the aspect of the temporal misalignment. Drawing inspiration from other areas, our paper introduces an early warning method applicable to energy systems, thereby filling a critical research gap.

So far, some research institutions, such as LANL, NREL and EMPA, have put forward theories related to energy security or resilience from different angles to assess or enhance the capacity of energy systems to sustain regular energy supply amidst disturbances. Grouped by disturbance sources, these theories can be classified as dealing with internal disruptions (e.g., gas leakage^{Error! Reference source not found.}) or external disruptions (e.g., extreme weather events^{Error! Reference source not found.}, ^{Error! Reference source not found.}, cyber attacks^{Error! Reference source not found.}). Organized chronologically, they focus on the ability of energy systems to prepare for, absorb, adapt to, and recover from

disruptive events before and after the disturbance^{Error! Reference source not found., Error! Reference source not found.}. In addition to traditional research on single energy resources, the exploration on the comprehensive resilience of coupled energy systems has also made breakthroughs (e.g., hurricane analysis^{Error! Reference source not found.}, disaster avoidance^{Error! Reference source not found.}). However, most of the above studies play a limited role in the prevention of gas-electric cascading failures for the lack of real-time coordination between the two energy entities when the gas network malfunction occurs. This is related to the temporal misalignment of the failure concerning the two energy systems: the failure has already transpired in the gas system, while the power system is still on the verge of the failure. Current theories about energy resilience are less prone to take the advantage of this propagation delay in the medium, which has actually been demonstrated by the earthquake early warning system in the area of geology and disaster response^{Error! Reference source not found., Error! Reference source not found.}.

Besides, to enhance the concreteness of the gas-electric early warning system, we have incorporated a preliminary overview of the general principles of early warning systems. This addition is intended to underscore that our gas-electric early warning system fundamentally embodies the application of established early warning technologies. This revision in the subsection *Early Warning of the Gas System* is shown as follows:

Early warning is defined as the perception of an impending danger. A comprehensive early warning system encompasses modules of sensing, analysis, decision-making, etc., intending to provide the response entity with more response time to minimize the losses^{Error! Reference source not found.}. In the area of disaster response, predicting the precise timing of disasters is challenging, but there exists correlation among them (e.g., an earthquake triggering a tsunami)^{Error! Reference source not found.}. This correlation involves a specific time interval surpassing the duration required by modules of the early warning system, rendering the early warning practicable in such instances. In coupled gas-electricity systems, there is such a correlation between the major failure of the gas network and the cascading failure of the power system caused by gas turbines. When the impacted gas turbines receive the control command from the EPCC, the cascading failure has not yet arrived at them.

Main Comments (#4):

- 4. Some of the events listed in Table 1 were so large and widespread that I doubt could be alleviated by the proposed method. Additionally, in some cases a failure of the grid could lead to the failure of electric powered gas pumps, which has happened in the past in New Mexico, causing further issues in the power grid. Please elaborate on how the proposed method would or would not help in historical events, such as the ones presented in Table 1.

Response:

Thank you for bringing this matter to our attention, and we have taken it into careful consideration.

In response to the reviewer's feedback and our own evaluation, we have opted to remove the contents of the original Table 1 in the Introduction section. This was partly because we found that some of the accidents in the original Table 1 (e.g., the event in London) did not fit our research topic. Our current approach involves highlighting the recurrent characteristic of gas-electric cascading failures through textual descriptions. This method offers a more flexible means of underscoring the widespread and unavoidable feature of such incidents.

However, compared to carbon emission reduction, the security of energy systems has not attracted enough attention from the public. With the expansion of gas networks, gas system malfunctions are now more likely to evolve into electric power outages. In February 2021, large-scale blackouts occurred in Texas and the south-central United States due to extremely cold weather^{Error! Reference source not found.}. Between February 8th and 20th, a gas generating

capacity of 106,585 MW encountered unplanned outages or derates, primarily stemming from issues in the gas system^{Error! Reference source not found.}. The resulting electric load shedding also caused outages to natural gas facilities, which in turn exacerbated the problem of an insufficient fuel supply to gas turbines. In the post-investigation report, gas-electric infrastructure interdependency was considered one of the most significant factors leading to the accident. In fact, similar incidents took place in the United States in 1983, 1989, 2003, 2004, 2006, 2008, 2009, 2010, and 2011 due to adverse weather conditions^{Error! Reference source not found.}. Additional factors such as pipeline leakage and illegal operations may also lead to gas-electric cascading failures, as exemplified in California (2015)^{Error! Reference source not found.} and Taiwan (2017)^{Error! Reference source not found.}. In short, power outages caused by gas network malfunctions are presently widespread and are anticipated to pose an increasing risk in the near future. Firstly, the transition from coal and oil to natural gas increases the dependence of electric power systems on gas networks, and even the latter is gradually replaced by decarbonized gas fuels, e.g., blue or green hydrogen or ammonia, gas networks persist^{Error! Reference source not found.}, ^{Error! Reference source not found.}. Secondly, an increasing number of studies indicate that extreme weather events have become more frequent than before^{Error! Reference source not found.}, ^{Error! Reference source not found.}, which may not only directly do harm to power systems but also threaten them by affecting the transportation of gas fuel^{Error! Reference source not found.}. Thirdly, global geopolitical tension amplifies the likelihood of human factors impacting the security of gas supplies, such as the Nord Stream gas pipeline explosion^{Error! Reference source not found.}.

Indeed, in the 2021 winter storm in Texas, the power supply loss led to the shutdown of some natural gas infrastructures such as gas pumps, thereby exacerbating the scale of the power outage. However, it is crucial to note that the basics of the *interactive failure* is still the failure of the gas system, which triggers the shutdown of the gas turbine, ultimately impacting the power supply. In other words, the blocking of gas-electric cascading failures is still the fundamental way to mitigate such interactive failures.

If we need to account for the impact of power outages on the natural gas infrastructure, it becomes necessary to intricately introduce the hierarchical load shedding method in the power system. However, such a model for this aspect is beyond the scope of our current paper, and obtaining relevant real-world data for such a model would be a formidable challenge. To underscore the innovation of the article, we mainly focus on the one-way propagation of the failure from the gas system to the power system, but the gas-electric early warning system remains applicable in such scenarios with bidirectional coupling between the gas system and the power system.

In fact, your suggestion has sparked a valuable idea for our future research endeavours. As well as electric powered gas pumps, the development of P2G (power to gas) units also strengthen the dependence of the gas system on the power system. This inspires us to explore bidirectional fault-blocking methods in future investigations.

REVIEWER COMMENTS

Reviewer #1 (Remarks to the Author):

Although authors have tried to address the proposed concerns still some drawbacks are present.

1- It is not true that pipeline is the best choice of transportation for fuels. Indeed it depends on the distance. You can refer to NREL evaluations for more explanation.

2- Provided case study are appreciated. However, it would be better to choose a case that highlights the advantages of this model and how it could prevent blackouts.

3- Alternative fuels and means of production or transportation have been developed to address such concerns. But literature review lacks a comprehensive review.

Reviewer #2 (Remarks to the Author):

Many of the raised concerns are addressed and the quality of the manuscript has improved significantly. However, there are certain aspects that are yet not very clear and addressed adequately. One of such domains is related to the concept of cascading failure that has been considered in this work. It seems that the authors have used it as a motivating concept but there is no clear definition of the cascade process or mechanism in neither of the gas or electric system in this work. In general, due to lack technical details the work seems to be more a review of data supporting the need for early warning system to electric grid due to malfunctions in gas network.

Moreover, the quality of the presentation can still be improved. For instance, Fig. 1 shows the data for 2020 but the caption says 2022. The captions are generally long and it is difficult to interpret the figures data.

Reviewer #3 (Remarks to the Author):

My comments have been answered properly.

Reviewer #4 (Remarks to the Author):

I appreciate the author's effort in substantially revising the paper. The new version addresses my concerns.

Response to the Reviewers: Reviewer 1

Comments to the Author:

- Although authors have tried to address the proposed concerns still some drawbacks are present.

Response:

We sincerely thank you for your careful review and valuable comments. We have revised the key concepts and reorganized the simulation results to make the merit and novelty of the work clearer. We hope that we have adequately addressed your concerns.

Main Comments (#1):

- 1. It is not true that pipeline is the best choice of transportation for fuels. Indeed it depends on the distance. You can refer to NREL evaluations for more explanation.

Response:

Thank you for your valuable suggestion.

A thorough examination of NERL reports on gaseous fuels and hydrogen energy has enriched our comprehension of the cost dynamics associated with different energy transportation modes, which dispelled some of our previous misconceptions. In the evolving landscape of future energy systems, with the growing integration of renewable generation, surplus electric energy will be converted into more easily storable chemical forms, e.g., methane or hydrogen (ref 1.1). These fuels can be transported through pipelines, trucks, rails, ships, etc. Generally, pipeline transportation costs are lower at long distances and large volumes (ref 1.2). At shorter distances or lower gas volumes, due to the high fixed costs involved in laying new lines and related ancillary equipment, pipelines become more expensive than trucks for inland transportation. Figure R1.1 illustrates the transport mode corresponding to the lowest cost for hydrogen energy (ref 1.3, where GH2 = gaseous hydrogen; LH2 = liquid hydrogen). In the early market "hydrogen success" scenario for FCEVs, depicted in Figure R1.2, both trucks and pipelines for hydrogen transportation will be constructed over the next few decades (ref 1.3). For a distance of 200 km, the comparative cost of hydrogen transportation is depicted in Figure R1.3 (ref 1.4). For fuel transport in coastal areas, ships can also compete with pipelines (ref 1.5).

Figure R1.1 Minimum delivery costs in SERA as a function of distance. (ref 1.3)

Figure R1.2 Geographic and capacity distribution of new hydrogen delivery capacity in the early market “hydrogen success” scenario, by transmission technology and capacity in 2050. (ref 1.3)

Figure R1.3 Cost of Hydrogen Transportation Alternatives for a distance of 200 km. (ref 1.4)

It can be seen that pipeline transportation is indeed not always the most economical way to transport alternative fuels. However, we also came across information on NERL's website regarding a research initiative named HyBlend. This project focuses on blending hydrogen into existing gas networks, which involves collaborative R&D efforts across four national laboratories. Leveraging the existing infrastructure of gas facilities allows for a more cost-effective increase in clean energy penetration compared to the construction of new hydrogen pipelines or trucking (ref 1.6). This aligns with the research projects in Germany and the UK highlighted in the Introduction part of our original manuscript. Consequently, in the forthcoming decades, the transition to decarbonized gas fuels will continue to rely on a sufficiently scaled gas network which serves as a good "energy storage" facility to enhance flexibility in the power system (ref 1.7). In essence, the coupling between gas networks and electric power systems will endure.

Based on the above contents, we have made slight revisions to the introduction of the article:

For electric power systems, it has become a common trend to gradually eliminate their dependence on fossil fuels and turn to low-carbon energy sources, in particular renewables such as wind and photovoltaics^{Error! Reference source not found., Error! Reference source not found.}. However, limited by the physical characteristic of power transmission and the intermittency of renewable generation, the development of conventional synchronous generators with sufficient capacity are still required that can be freely adjusted to maintain the balance between generation and the load of power systems^{Error! Reference source not found., Error! Reference source not found.}. Among all types of conventional generators, gas turbines have attracted public attention due to their superior performance and lower prices^{Error! Reference source not found.}. In contrast to coal- and oil-fired generators, gas turbines emit lower levels of carbon and pollutants and exhibit higher ramp rates which facilitates the integration of variable renewable resources^{Error! Reference source not found., Error! Reference source not found., Error! Reference source not found.}. Many countries have built or are building large-scale gas networks and gas-fired power generators to reduce their carbon emission intensity^{Error! Reference source not found.}, as shown in Figure 1.

Moving towards carbon neutrality, although the use of natural gas as a fossil fuel will eventually be reduced, the gas infrastructure will be repurposed as an important energy carrier of alternative fuels (e.g., green hydrogen) with storage and transmission functionality^{Error! Reference source not found., Error! Reference source not found.}. Although the share of hydrogen production from electrolysis of water is not as high as that from fossil fuels at present, the former is shown to be the best way to obtain large amounts of sustainable hydrogen with neither pollutants nor carbon emission^{Error! Reference source not found., Error! Reference source not found.}. Moreover, transporting these alternative fuels through established gas networks has proved to be a more economical way than trucking or constructing new pipelines^{Error! Reference source not found.}. In early 2021, Shell, Mitsubishi Heavy Industries, Vattenwall and Wärme Hamburg signed a letter of intent to develop renewable hydrogen production in Hamburg, where existing gas networks would be expanded to accommodate the transport of hydrogen^{Error! Reference source not found.}. In the same year, an initiative called HyBlend was launched by the Office of Energy Efficiency and Renewable Energy (EERE) of the US, including R&D conducted by four national laboratories^{Error! Reference source not found.}. In July 2023, UK Research and Innovation (UKRI) declared a funding allocation of £95.3 million to push forward 10 large-scale demonstration projects, aiming to investigate the feasibility of blending green hydrogen into existing gas networks^{Error! Reference source not found.}. In this way, when the generation capacity of renewables surpasses the load demand, the surplus power can be converted into green hydrogen and injected into the gas system; when the renewable generation falls short of the load demand, the power deficit can be supplemented through gas turbines.

References:

- 1.1 Topolski, Kevin, et al. "Hydrogen blending into natural gas pipeline infrastructure: review of the State of technology." (2022).
- 1.2 Melaina, Marc W., Olga Antonia, and Michael Penev. "Blending hydrogen into natural gas pipeline networks: a review of key issues." (2013).
- 1.3 Melaina, M. SERA Scenarios of Early Market Fuel Cell Electric Vehicle Introductions: Modeling Framework, Regional Markets, and Station Clustering; NREL (National Renewable Energy Laboratory). No. NREL/PR-5400-64395. National Renewable Energy Lab. (NREL), Golden, CO (United States), 2015.
- 1.4 Reddy, B. Sudhakara, and P. Balachandra. "Hydrogen energy for Indian transport sector: A Well-to-wheel techno-economic and environmental feasibility analysis." (2012).
- 1.5 Brogan, James J., et al. "Transportation Energy Futures Series: Freight Transportation Modal Shares: Scenarios for a Low-Carbon Future." (2013).

1.6 Topolski, Kevin, et al. Techno-Economic Analysis of Blending Hydrogen into Natural Gas Transmission Networks. No. NREL/PR-5400-85123. National Renewable Energy Lab. (NREL), Golden, CO (United States), 2023.

1.7 Melaina, M., and J. Eichman. Hydrogen energy storage: Grid and transportation services (Technical report). No. NREL/TP-62518. National Renewable Energy Lab. (NREL), Golden, CO (United States), 2015.

Main Comments (#2):

- 2. *Provided case study are appreciated. However, it would be better to choose a case that highlights the advantages of this model and how it could prevent blackouts.*

Response:

Thank you for your valuable suggestion.

We have modified the provincial example of the gas-electric early warning system. Firstly, the description of all gas-fired generators in the power system has been added to the figure, and the SAET of impacted gas turbines is marked to reflect the emergency degree of different failures. Secondly, we have added wind power failures, which makes our simulation closer to the reality of winter storms. Thirdly, the dynamic proactive control method has been used to further reduce the power deficit and highlight the advantages of the gas-electric early warning system. Concretely, the modifications are as follows:

In order to verify the scalability of the gas-electric early warning system, a case study of the provincial system is also conducted as shown in Figure 7. The provincial gas network adopts the structure of multi-end gas sources, which effectively reduces the impact of a single major failure on the power system. Therefore, we simulate simultaneous failures of multiple gas sources and gas pipelines during a winter storm, as well as malfunctions of two wind farms in the power system. According to the topological analysis of the gas network, six out of twelve gas-fired generators in the power system will lose stable gas supply when the failures occur, and their SAET ranges from minutes to hours depending on the distance between the gas turbine and the failure location. We present generation curves of typical generators in the power system including gas turbines, impacted gas turbines, coal-fired steam turbines, and nuclear power plants, both in the static proactive control strategy and the dynamic proactive control strategy. Among all types of normal generators, nuclear power plants are set to bear the base load, so they are not involved in the control process. Compared to swift adjustment of gas turbines, the generation ramp rate of coal-fired steam turbines has a stricter limit due to their complex physical structures. It is demonstrated that, in contrast to passive control which leads to a maximum power deficit of 1,717.5 MW and a total energy deficit of 367.9 MWh, the maximum power deficit and the total energy deficit in the static and dynamic proactive control strategies are 1,074.9 MW, 215.0 MWh and 664.7 MW, 14.0 MWh, respectively.

Fig. 7 | Provincial demonstration of the gas-electric early warning system. a, Malfunctions in the coupled gas-electricity system. When large-scale failures occur in the gas system, certain gas turbines are impacted and their SAET values are marked nearby. b, Generation curves of impacted gas turbines (GT7~GT12) in proactive control strategies. Compared to the static proactive control strategy, the dynamic proactive control strategy provides a longer power regulation time for impacted gas turbines. c, Generation curves of normal generators (GT1~GT6, NP1, NP2, ST1, ST2) in proactive control strategies. Typically, gas turbines have greater regulatory capabilities compared to coal-fired steam turbines, while nuclear power plants are rarely involved in the generation adjustment process.

Furthermore, in order to underscore the connection between power deficits and power outages, we have provided additional technical explanations in the "Proactive Control of the Power System" section. Through the reduction of power deficits, the gas-electric early warning system can effectively minimize the occurrence of power outages originated from malfunctions in the gas system.

Main Comments (#3):

- 3. Alternative fuels and means of production or transportation have been developed to address such concerns. But literature review lacks a comprehensive review.

Response:

Thank you for your suggestions.

Classic literature reviews about the production, storage, and transportation of alternative fuels (e.g., hydrogen) have been documented in many professional journals (ref 3.1~3.3). Concerning production, prevalent engineering techniques include steam reforming, biomass gasification, and water electrolysis. Concerning storage, conventional methods encompass compressed gas, liquefaction, metal hydride, and metal-organic frameworks. Concerning transportation, research has identified viable modes of transportation and conducted cost comparisons under various hydrogen storage forms.

In essence, the foundation of the gas-electric early warning system discussed in this paper is the reliance of the power system on the gas network. So far, natural gas has gained prominence in many countries, especially in China, owing to its cleaner attributes compared to coal and oil, which increases the dependence of the power system on the gas system on a world-average basis (as reflected in Figure 1 of the manuscript). Besides, the evolution of alternative fuels such as hydrogen and methane, along with their utilization in pipeline transportation (either transported independently or injected into the existing natural gas infrastructures), will sustain the integration of the gas network and the power system for an extended

duration even in the stage where natural gas as a fossil fuel gradually gives way to cleaner alternatives.

Therefore, we think that a detailed review of the production and transmission of alternative fuels (hydrogen and methane) would not be essential. Instead, the reasons why gas networks will persist and remain coupled to the power system could be more emphasized, which was partly answered in reply #1. So far, the research on blending hydrogen into existing gas networks has been supported by many engineering initiatives across different countries (ref 3.4~3.6). Some key issues in these studies (including green hydrogen production and transmission) are described in the manuscript to support our claim that the dependence of the power system on the gas network will persist.

Therefore, we have made modifications and additions to the relevant paragraph in the Introduction section to reveal the relationship between alternative fuels and renewable energy, which has been shown in reply #1.

References:

- 3.1 Qureshi, Fazil, Mohammad Yusuf, and Bawadi Abdullah. "A brief review on hydrogen production to utilization techniques." 2021 third international sustainability and resilience conference: climate change. IEEE, 2021.
 - 3.2 Zhong, Zhiyao, et al. "Power-to-hydrogen by electrolysis in carbon neutrality: Technology overview and future development." CSEE Journal of Power and Energy Systems (2023).
 - 3.3 Ursua, Alfredo, Luis M. Gandia, and Pablo Sanchis. "Hydrogen production from water electrolysis: current status and future trends." Proceedings of the IEEE 100.2 (2011): 410-426.
 - 3.4 Shell, Mitsubishi Heavy Industries. Vattenfall and Wärme Hamburg Sign Letter of Intent for 100MW Hydrogen Project in Hamburg (2021).
 - 3.5 UK Research and Innovation. Ofgem Strategic Innovation Fund announces 10 funded projects (2023).
 - 3.6 Topolski, Kevin, et al. Techno-Economic Analysis of Blending Hydrogen into Natural Gas Transmission Networks. No. NREL/PR-5400-85123. National Renewable Energy Lab. (NREL), Golden, CO (United States), 2023.
-

Response to the Reviewers: Reviewer 2

Comments to the Author:

- *Many of the raised concerns are addressed and the quality of the manuscript has improved significantly. However, there are certain aspects that are yet not very clear and addressed adequately.*

Response:

We sincerely thank you for your careful review and the valuable suggestions to improve the quality of the paper! According to your suggestions, we have made corresponding changes to the manuscript. The following are the answers to your questions and concerns.

Main Comments (#1):

- *1. One of such domains is related to the concept of **cascading failure** that has been considered in this work. It seems that the authors have used it as a motivating concept but there is no clear definition of the cascade process or mechanism in neither of the gas or electric system in this work. In general, due to lack **technical details** the work seems to be more a review of data supporting the need for early warning system to electric grid due to malfunctions in gas network.*

Response:

Thank you again for your valuable comment.

Numerous professional studies have focused on the propagation characteristics of cascading failures within power transmission systems. For example, transmission lines are typically equipped with relay protection devices which cut off the line to prevent overload. The power redistribution induced by the failure of key transmission lines may lead to transmission line overloads and failures. This, in turn, results in a new cycle of power redistribution, exacerbating the operational state of the system. Common research approaches encompass methodologies based on system network topology (ref 1.1~1.2) and those based on power simulation (ref 1.3~1.4). For another example, with the penetration of renewable energy, instances of renewable off-grid events triggered by cascading failures are becoming more prevalent, e.g., both too fast frequency drop and long-time low voltage will cause renewable energy off-grid. In power systems characterized by a substantial presence of power electronic equipment, the consequence of voltage and frequency instability is also notably more severe compared to that in traditional power systems. There are also many relevant studies which handle such cascading failures (ref 1.5~1.6). Given the swift propagation of electromagnetic fields at the speed of light and the rapid response time of relay protection devices, cascading failures in power systems demand urgent attention, which underscores the imperative for effective decision-making by the EPCC and the system's emergency control capabilities.

In contrast, research on cascading failures within gas systems remains comparatively limited, where existing studies primarily focuses on the impact of major failures (e.g., pipeline leakages) on the pressure at key nodes of the network (ref 1.7). This is mainly attributed to the fact that gas supply disruptions generally do not have the same social impact as power supply disruptions. Additionally, the gas system exhibits slow dynamics, with failure propagation occurring at a considerably slower pace than that observed in power systems. Consequently, the GDC typically has more time on failure resolution to reduce the losses. Nevertheless, with the growing integration of various energy sources, the propagation of gas system failures to other energy systems continues to rise. Research on the integrated energy system has in turn promoted the discussion of the mechanism of cascading failures within the gas network.

So far, the mechanism of cross-energy cascading failure in integrated electricity and gas systems (IEGSs) has become a new research hotspot (ref 1.8~1.10). These cascading failures are generally facilitated by coupling components, such as gas turbines, which mediate failure propagation from the gas network to the power system, and electrically driven compressors or power to gas (P2G) units, which mediate failure propagation from the power system back to the gas network. Some studies also direct their attention towards the bi-directional propagation and re-entry processes of cascading failures, which aims to provide a more comprehensive description of the consequences associated with cascading failures in the IEGS. In this work, our primary focus lies on the propagation process of cascading failures originated from the gas network to the power system, along with the exploration of the mitigation method. We mainly develop the early warning and proactive control method, leveraging the disparity in the time scales of dynamic processes of the two systems. To implement this strategy, we have selected the common gas turbine as the coupling component and modelled the power system and the gas system in a quasi-steady and a dynamic state, respectively.

In summary, incorporating your valuable comments and considering the public interest of the article, we have involved typical pathways of cascading failures within gas systems and power systems. For the gas system, pressure deviation stands out as the primary driver of cascading failures. Consequently, in the section "Early Warning of the Gas System", we have underscored the role of pressure in cascading failures of the gas network, and emphasized that its slow change characteristic is the key guarantee for blocking such cascading failures. For the power system, the reason for cascading failures can be very complex, so we have chosen the frequency drops that are most associated with active power deficits emphasized in the paper, and highlight the impact of increased renewable energy penetration. The corresponding text in the section "Proactive Control of the Power System" has been revised.

Concretely, the additions and modifications are shown as follows:

Early Warning of the Gas System

Early warning is defined as the perception of an impending danger. A practical early warning system encompasses modules of sensing, analysis, decision-making, etc., intending to provide the management entity with more response time to minimize the losses^{Error! Reference source not found.}. For example, in the area of disaster response, predicting the precise timing of a tsunami is challenging, but there exists obvious correlation between a coastal earthquake and a tsunami^{Error! Reference source not found.}. This correlation involves a time interval which exceeds the reaction time required for the public to prepare for the tsunami, making the tsunami early warning system practicable.

In coupled gas-electricity systems, there is such a correlation between certain major failures of the gas network and cascading failures of the electric power system. When the information is limited to the power system, it is hard for the EPCC to predict unplanned outages of gas-fired generators. However, when the object of concern expands to the coupled gas-electricity system, there is a clear sequential correlation between gas supply interruptions and their impact on gas turbines due to insufficient inlet pressure. For gas-fired generators, when the inlet pressure drops to a certain extent, their maximum power generation is greatly reduced for safety purposes; as the inlet pressure drops further, they must be shut down to prevent equipment anomalies. In fact, pressure is one of the most crucial state variables for the normal operation of gas facilities including valves, compressors and gas loads. Its deviation commonly serves as a reason for cascading failures within the gas system, just as deviations of voltage lead to cascading failures within the power system. Nevertheless, a comparison between the pipeline equations of gas systems and the telegraph equations of power systems reveals a notable distinction. Gas systems demonstrate a stronger capacity to maintain constant pressure compared to the capability of power systems to maintain constant

voltage. Based on typical gas network and gas turbine parameters, it takes levels of minutes for the inlet pressure of the gas turbine to drop to the protection threshold, assuming a complete interruption of the gas supply several kilometres away from the gas turbine. In contrast, it only takes levels of seconds for the gas-electric early warning system to summarize the information about the gas failure and initiate the control process of the power system. Therefore, when power generators receive the control command from the EPCC, cascading failures have not arrived at the power system, ensuring the feasibility of the gas-electric early warning system.

Proactive Control of the Power System

The electric power system requires a real-time balance between generation and the load, which will be broken when the impacted gas turbines are forced to reduce their power generation. At this time, there exists a difference between the electric load power and the actual power generation, defined as the *power deficit*. Generally, the EPCC must conduct emergency control to compensate for the power deficit; otherwise, the frequency of the power system starts to drop. Common emergency control methods involve utilizing the rotating reserve capacity of generators and implementing low-frequency load shedding, and the latter is the primary reason for power outages. Modern power systems typically adhere to the N-1 principle, ensuring that the withdrawal of any single component does not result in power outages. However, as the penetration of renewables increases, the frequency inertia of the power system gradually diminishes, which means that the power deficit will lead to a faster decline in the system frequency. In such cases, even the outage of a single generator may develop into cascading failures which involve the withdrawal of other generators, eventually necessitating low-frequency load shedding by the EPCC^{Error! Reference source not found., Error! Reference source not found.}. Hence, when a major failure occurs in the gas network, the gas-electric early warning system must find ways to reduce the power deficit arising from the impacted gas turbines.

When information is limited to the power system, it is difficult for the EPCC to adjust the generation of the impacted gas turbines in advance, which results in the direct impact of insufficient inlet pressure of the gas turbine on the power system. In certain instances, although some gas turbines reported the situations of gas supply disruptions to the EPCC, the reports from different gas turbines were not systematic, leading to the improper utilization of rotating reserve capacities. Once the gas-electric early warning system is introduced, the EPCC can obtain an overview of the gas system failure in advance and formulate control strategies before the impact of the failure on the power system is experienced, which we refer to as *proactive control* of the power system. On the contrary, if the EPCC adjusts the output of generators only when the power deficit actually occurs, we refer to such a strategy as *passive control* of the power system. In the proactive control strategy, each normal generator with reserve capacity has a longer time to compensate for the power generation reduction of the impacted gas turbine, leading to a substantial reduction or even elimination of the power deficit, as shown in Figure 4. In certain cases, the ALP may not be sufficient to entirely support the substitution of the impacted gas turbine by other generators. If this happens, a power deficit persists, but it is significantly reduced compared to that in the passive control strategy.

References:

- 1.1 Watts, Duncan J., and Steven H. Strogatz. "Collective dynamics of 'small-world' networks." *nature* 393.6684 (1998): 440-442.
- 1.2 Barabási, Albert-László, and Réka Albert. "Emergence of scaling in random networks." *science* 286.5439 (1999): 509-512.

-
- 1.3 Ni, Ming, et al. "Online risk-based security assessment." IEEE Transactions on Power Systems 18.1 (2003): 258-265.
 - 1.4 Salmeron, Javier, Kevin Wood, and Ross Baldick. "Analysis of electric grid security under terrorist threat." IEEE Transactions on power systems 19.2 (2004): 905-912.
 - 1.5 Shen, Zhengwei, et al. "Analysis of cascading failure evolution process of high proportion new energy power system." Journal of Physics: Conference Series. Vol. 2276. No. 1. IOP Publishing, 2022.
 - 1.6 Wang, Leibao, Bo Hu, and Kaigui Xie. "Dynamic Cascading Failure Model for Blackout Risk Assessment of Power System With Renewable Energy." E3S Web of Conferences. Vol. 257. EDP Sciences, 2021.
 - 1.7 Wang, Y., H. Huang, and B. N. Su. "Vulnerability analysis of urban natural gas network under cascading failures." 2015 International Conference on Industrial Technology and Management Science. Atlantis Press, 2015.
 - 1.8 Li, Shuai, et al. "A machine learning-based vulnerability analysis for cascading failures of integrated power-gas systems." IEEE Transactions on power systems 37.3 (2021): 2259-2270.
 - 1.9 Bao, Zhejing, et al. "Cascading failure propagation simulation in integrated electricity and natural gas systems." Journal of Modern Power Systems and Clean Energy 8.5 (2020): 961-970.
 - 1.10 Bao, Zhejing, Zhewei Jiang, and Lei Wu. "Evaluation of bi-directional cascading failure propagation in integrated electricity-natural gas system." International Journal of Electrical Power & Energy Systems 121 (2020): 106045.
-

Main Comments (#2):

- 2. Moreover, the quality of the presentation can still be improved. For instance, Fig. 1 shows the data for 2020 but the caption says 2022. The captions are generally long and it is difficult to interpret the figures data.

Response:

Thank you for your comment.

At present, we have modified Figure 1 to make the year more clearly marked. Some changes have also been made to the caption of Figure 1 to highlight the core content of the figure. The modifications are as follows:

Fig. 1 | Development of natural gas utilities worldwide. a, Kilometres of gas pipelines worldwide at the end of 2022. The size of each sector on the left indicates the operating length of gas pipelines in a region, while that

on the right denotes the length of gas pipelines in development, reflecting the growth momentum of gas infrastructure. **b, Global power generation and the proportion of natural gas in 2010 and 2020.** The size of each sector indicates the annual power generation in a region, while the dashed area denotes the proportion generated by natural gas. Data are retrieved from the literature Error! Reference source not found.. Error! Reference source not found..

Moreover, we have made slight changes to the captions of Figure 3~5, aiming to highlight the point of our presentations. Some language expressions have also been modified in the manuscript where the text is highlighted in yellow.

REVIEWERS' COMMENTS

Reviewer #1 (Remarks to the Author):

All my concerns are addressed.

Reviewer #2 (Remarks to the Author):

The reviewer still believes that the depth of the presented work in the domain of cascading failures is very limited. This work does not fit in the cascading failure analysis scope with its lack of cascade model definition and model. It seems that this work is relying on data to highlight some aspects of interdependency between these two systems, which is valuable, but not very technical.

Response to the Reviewers: Reviewer 2

Comments to the Author:

- *The reviewer still believes that the depth of the presented work in the domain of cascading failures is very limited. This work does not fit in the cascading failure analysis scope with its lack of cascade model definition and model. It seems that this work is relying on data to highlight some aspects of interdependency between these two systems, which is valuable, but not very technical.*

Response:

We appreciate the reviewer's feedback and acknowledge the concern regarding the depth of our work in the domain of cascading failures. We understand the importance of a comprehensive cascade model in analysing cascading failures in single energy systems, and we recognize that our current work may not fully meet the expectations in this regard.

Our intention in this paper was to primarily focus on highlighting the interdependency between the power system and the gas network, particularly in the context of failure mitigation at their interface. In the proposed gas-electric early warning method, there is only information exchange at the boundary between the gas system and the power system, satisfying principles of privacy protection and not necessitating detailed cascade models within single energy systems. While we agree that our approach relies on data analysis to illustrate these interdependencies, we believe that this perspective provides valuable insights into the operational challenges faced by integrated energy systems (e.g., coupled gas-electricity system).

In fact, we have noticed that in professional journals, the characteristics of cascading failures within single energy systems (especially the power system) have been extensively studied. In contrast, there has been insufficient research on the propagation characteristics and mitigation methods of cross-energy cascading failures between gas and power systems. Constructing a comprehensive model that encompasses the development of faults within the gas system, their propagation from the gas system to the power system, and their subsequent development within the power system would require significant additional efforts. These efforts, indeed, are not the primary focus of our study, which may also lead to exceeding the prescribed word limit for the article.

However, we acknowledge the reviewer's point that a more detailed cascade model would enhance the technical rigor of our work. In our future research, we plan to address this gap by incorporating a more comprehensive cascade model into our analysis. By integrating this model with the early warning approach proposed in our work, we aim to provide a more robust and technical framework for analysing cascading failures in integrated energy systems. In response, we have modified the Discussion as follows:

During the energy transition towards carbon neutrality when renewable energy cannot become the main power source, gas turbines will replace many traditional coal-fired power generation sources because of their lower carbon emissions and higher control speed. In the future, when renewable energy can be applied on a large scale, owing to the development of hydrogen energy, gas pipelines and gas storage facilities will be maintained and be repurposed as important energy storage and transmission components. Due to the power system's dependence on the gas network, major failures in the gas system may lead to certain gas turbines shut down, resulting in a significant power deficit in the power system, defined as gas-electric cascading failures. In such scenarios, traditional theories of power system security, especially those based on the N-1 principle, face limitations. Real-world experiences of power outages highlight that extreme external events can disrupt fuel supply to multiple gas turbines, rendering security protection measures based on the N-1 principle ineffective. In response, we draw inspiration from the early warning principles successfully applied in the disaster response, and propose the gas-

electric early warning system. This innovative system offers extended time for adjusting generator outputs, aiming to minimize the emergency control cost and reduce the likelihood of power outages.

In the gas-electric early warning system, the GDC analyses global information regarding gas major failures and notifies the EPCC in advance, so that the latter can systematically optimize the operation of the power system. The quantification of this global information is achieved through early warning indicators including the AET and the ALP. In our study, these two indicators can provide sufficient information for the proactive control of the power system, eliminating the need for the EPCC to access more detailed information about the gas system. This underscores that the gas-electric early warning system does not simply combine the information of the two systems; it meets the demand of privacy protection across multiple entities. Based on the early warning indicators, the EPCC can perform proactive control instead of passive control on the power system. Compared to the passive control strategy, the proactive control strategy effectively reduces the power deficit caused by major failures in the gas network. The formulation of proactive control strategies can be modelled as a linear programming problem with unknown control times of the impacted gas turbines. By dynamically extending these control times, a larger ALP can be used to minimize the power generation deficit. In the case study of a province in East China, the power generation deficit can be reduced for different major failures in the gas network. In extreme cases, multiple failures may occur in the gas network at the same time, and the role of the gas-electric early warning method will be more significant.

The basic premise of the gas-electric early warning system is that when a major failure occurs in the gas system, the storage capacity of gas pipelines serves as a buffer for failure propagation. If this buffer is not enough to maintain the inlet pressure, the effectiveness of the early warning system will be limited. In practical engineering applications, it is possible to configure typical failure scenarios within the gas system by considering the failure probabilities of various components. This assessment serves to evaluate whether the storage capacity aligns with the requirements of the gas-electric early warning system. If inadequacies are identified, additional gas storage facilities can be installed to further ensure the feasibility of the gas-electric early warning system, thereby enhancing the resilience of coupled gas-electric system to external extreme events.

In this paper, we mainly focus on exploring the interdependency between the power system and the gas network, and propose an innovative method to mitigate gas-electric cascading failures at the interface of the two systems. In future research, the early warning approach presented in this paper can be integrated with more detailed cascading failure models, thus yielding a more technical and comprehensive framework to address such problems. Moreover, in future energy systems, electricity, heat, gas, transportation and other forms of energy will exhibit further coupling relationships, while their physical characteristics vary^{Error! Reference source not found.}. As a result, the cross-energy early warning ideas demonstrated in this paper can be extended to other energy flow combinations in future work, e.g., the coupled transportation-electricity system.

Once again, we thank the reviewer for their valuable feedback, and we are committed to addressing their concerns in our future research efforts.